

# Synoptic meteorological modes of variability for fine particulate matter (PM$_{2.5}$) air quality in major metropolitan regions of China

Danny M. Leung[1], Amos P. K. Tai[1,2], Loretta J. Mickley[3,4], Jonathan M. Moch[4], Aaron van Donkelaar[5], Lu Shen[3], Randall V. Martin[5,6]

[1]Earth System Science Programme and Graduate Division of Earth and Atmospheric Sciences, The Chinese University of Hong Kong, Hong Kong
[2]Institute of Environment, Energy and Sustainability, The Chinese University of Hong Kong, Hong Kong
[3]School of Engineering and Applied Sciences, Harvard University, Cambridge, Massachusetts, USA
[4]Department of Earth and Planetary Sciences, Harvard University, Cambridge, Massachusetts, USA
[5]Department of Physics and Atmospheric Science, Dalhousie University, Halifax, N. S. Canada
[6]Harvard-Smithsonian Center for Astrophysics, Cambridge, Massachusetts, United States

*Correspondence to*: Amos P. K. Tai (amostai@cuhk.edu.hk)

**Abstract.** In this study, we use a combination of multivariate statistical methods to understand the relationships of PM$_{2.5}$ with local meteorology and synoptic weather patterns in different regions of

China across various timescales. Using June 2014 to May 2017 daily total PM$_{2.5}$ observations from ~1500 monitors, all deseasonalized and detrended to focus on synoptic-scale variations, we find strong correlations of daily PM$_{2.5}$ with all selected meteorological variables (e.g., positive correlation with temperature but negative correlation with sea-level pressure throughout China; positive and negative correlation with relative humidity in northern and southern China, respectively). The spatial patterns

suggest that the apparent correlations with individual meteorological variables may arise from common association with synoptic systems. Based on a principal component analysis on 1998–2017 meteorological data to diagnose distinct meteorological modes that dominate synoptic weather in four major regions of China, we find strong correlations of PM$_{2.5}$ with several synoptic modes that explain 10% to 40% of daily PM$_{2.5}$ variability. These modes include monsoonal flows and cold frontal passages

in northern and central China associated with the Siberian high, onshore flows in eastern China, and frontal rainstorms in southern China. Using the Beijing-Tianjin-Hebei (BTH) region as a case study, we further find strong interannual correlations of regionally averaged satellite-derived annual mean PM$_{2.5}$ with annual mean relative humidity (positive) and springtime fluctuation frequency of the Siberian high (negative). We apply the resulting PM$_{2.5}$-to-climate sensitivities to IPCC Coupled Model

Intercomparison Project Phase 5 (CMIP5) climate projections to predict future PM$_{2.5}$ by 2050s due to climate change, and find a modest decrease of ~0.1 μg m$^{-3}$ in annual mean PM$_{2.5}$ in the BTH region, which represents the compensating effects of enhanced relative humidity and synoptic frequency.





## 1. Introduction

Air pollution caused by high surface concentrations of particulate matter (PM) and ozone in megacities are of utmost public health concern in China nowadays (Xu et al., 2013). Outdoor air pollution in China alone has been attributed to over 1 million premature deaths every year (Cohen et al., 2017).

Many epidemiological studies have documented the harmful effects of fine particulate matter ($PM_{2.5}$, particles with an aerodynamic diameter of or less than 2.5 μm) in particular on cardiovascular and respiratory health (Cao et al., 2012; Krewski et al., 2009; Madaniyazi et al., 2015; Pope and Dockery, 2006). The severity of $PM_{2.5}$ pollution is known to be strongly dependent not only on emissions but also on weather conditions. For example, Zhang et al. (2016) showed using GEOS-Chem that cold surge

occurrences over northern China attribute to about half of the variability of total $PM_{2.5}$. Several modeling studies have examined the effects of historical (Fu et al., 2016) and future (Jiang et al., 2013) changes in emissions and climate (i.e., long-term changes in weather statistics) on $PM_{2.5}$ air quality in East Asia, but large uncertainty remains due to the complex $PM_{2.5}$-meteorology interactions (Jiang et al., 2013; Shen et al., 2017; Tai et al., 2012b). Such poor understanding stems mainly from the diverse sensitivities of

different $PM_{2.5}$ chemical components to meteorological changes, and from the strong coupling of $PM_{2.5}$ with synoptic circulation and the hydrological cycle. In this study, we apply a combination of multivariate statistical techniques to identify important local-scale meteorological variables and synoptic-scale meteorological modes that dominantly control the daily and interannual variability of $PM_{2.5}$ in China, and illustrate how these modes enable effective diagnosis of the effects of future synoptic circulation changes

on China $PM_{2.5}$ air quality.

China has experienced deteriorating air quality since the 1990s due to rapid industrial and economic developments. Haze and smog pollutions with dangerous levels of $PM_{2.5}$ are becoming more common in the most developed and highly populated city clusters in China (Chan et al., 2008; Zhang et al., 2007; Zhang et al., 2014). For example, Beijing's annual mean $PM_{2.5}$ concentration increased

dramatically from 12 μg m$^{-3}$ in 1973 to 66 μg m$^{-3}$ in 2013, with an average increase rate of 0.7 μg m$^{-3}$ yr$^{-1}$ in the past four decades (Han et al., 2016). Urban $PM_{2.5}$ originates from many sources including power plant, industry, vehicular emissions, road and soil dust, biomass burning, and agriculture activities (Zhang et al., 2015), but the regional concentrations are also strongly dependent on pan-regional transport (e.g., Jiang et al., 2013) and ventilation by atmospheric circulation (e.g., Chen et al., 2008; Zhang et al., 2012;

Zhang et al., 2016).

Local meteorological conditions are known to strongly influence the levels of all air pollutants including $PM_{2.5}$. $PM_{2.5}$-meteorology interaction is complex due to the varying responses of $PM_{2.5}$ species to different meteorological variables. Higher temperature favors the formation of sulfate and secondary organic aerosols due to the faster oxidation of sulfur dioxide ($SO_2$) and volatile organic compounds

(VOCs) (Jacob and Winner, 2009). Higher temperature also increases the emissions of biogenic VOCs



from vegetation, especially in southern China where high-emitting broadleaf evergreen trees are prevalent (Ding et al., 2012; Zhang and Cao., 2015). Higher temperature favors the volatilization of nitrate, ammonium and semivolatile organics by shifting the gas-aerosol phase equilibria toward the gas phase (Jiang et al., 2013; Shen et al., 2017), thereby decreasing these components. Depending on the region, an

increase in relative humidity (RH) may enhance the production of hydroxyl (OH) radical and hydrogen peroxide ($H_2O_2$), which promotes $SO_2$ oxidization and increases the uptake of semivolatile components including nitrate and organics (Seinfeld and Pandis, 2016). Precipitation, via its direct scavenging effect, is a principal sink for all $PM_{2.5}$ components (Koch et al., 2003; Tai et al., 2010). Meanwhile, both strong wind and boundary layer mixing also tend to ventilate or dilute $PM_{2.5}$ (Chen et al., 2008; Jacob and Winner,

2009; Wang et al., 2012; Zhang and Cao, 2015). For instance, Han et al. (2016) found that annual mean $PM_{2.5}$ in Beijing was negatively correlated with annual mean wind speed over 1973–2013, illustrating the importance of ventilation on interannual $PM_{2.5}$ variability.

In addition to local meteorological conditions, synoptic-scale circulation patterns also play important roles in driving $PM_{2.5}$ variability. Different classification schemes for a wide range of synoptic

circulation patterns have been researched extensively (Huth et al., 2008), and used worldwide to evaluate pollution-meteorology interactions (e.g., McGregor and Bamzelis, 1995; Shahgedanva et al., 1998; Shen et al., 2017; Tai et al., 2012a; Zhang et al., 2012). Tai et al. (2012a) showed that cold fronts associated with midlatitude cyclone passages and maritime inflows were the major ventilation mechanisms of $PM_{2.5}$ in the US. Shen et al. (2017) further showed that variability of $PM_{2.5}$ over the US explained by both local

meteorology and synoptic factors (43%) are in average about 10% higher than solely using local meteorology (34%). In Asia, Chen et al. (2008) demonstrated that synoptic high-pressure systems in northern Mongolia associated with cold fronts facilitate the dispersion of air pollutants over northern China, whereas a surface high centered on BTH favors accumulation. Zhang et al. (2013) showed similar results by extracting nine distinct synoptic pressure patterns at the North China Plain (NCP), and

discovered that weak pressure tendency in NCP favor pollutant accumulation. Zhang et al. (2016) found that a cold surge associated with the East Asian winter monsoon significantly reduced $PM_{2.5}$ concentration in Beijing by 110 μg m$^{-3}$ within a few days. Moreover, the effects of local meteorology and synoptic circulation are not independent of each other. For instance, Tai et al. (2012a) found that much of the apparent observed correlation of $PM_{2.5}$ with temperature and pressure in the eastern US are attributable

to common association with cold frontal passages. To understand how meteorological changes may affect future $PM_{2.5}$ air quality, therefore, requires keen consideration of the covariation of meteorological variables with synoptic-scale phenomena in an integrated framework (Jiang et al., 2005).

In this study, we perform correlation analysis to estimate the sensitivities of observed daily total $PM_{2.5}$ to a suite of local meteorological variables from June 2014 to May 2017. As we will see in Sect. 3,

however, correlations between local meteorology and $PM_{2.5}$ are complicated by covariations among individual meteorological variables, which are at least partially driven by synoptic systems. We therefore



apply principal component analysis to construct different meteorological modes that distinguish between unique synoptic-scale meteorological regimes, and principal component regression of daily $PM_{2.5}$ on these modes to not only interpret the observed correlations of daily $PM_{2.5}$ with individual meteorological variables, but also determine the dominant meteorological modes of daily $PM_{2.5}$ variability, in four major

city clusters of China: the Beijing-Tianjin-Hebei (BTH), the Yangtze River Delta (YRD), the Pearl River Delta (PRD), and the Sichuan Basin (SCB) (Fig. 1). Furthermore, using BTH as a case study, we apply a spectral analysis on the time series of dominant meteorological modes over the past decade to examine the interannual correlations between synoptic frequencies and annual mean $PM_{2.5}$. We finally construct a statistical model using annual median synoptic frequency and annual mean local meteorology to project

2000–2050 $PM_{2.5}$ changes, given present-day and future climate simulations by an ensemble of climate models. This study represents an advancement over that of Tai et al. (2012a, b) in terms of methodology by considering the joint effects of synoptic frequency and local meteorology, and is on a par with the work of Shen et al. (2017), who focused on the US instead. Our work represents the first attempt to apply these methods to China air quality in an effort to derive a statistical projection of future $PM_{2.5}$

concentrations based on historical $PM_{2.5}$-meteorology relationships. These historical relationships can also be used to compare results from process-based models (e.g., Jiang et al., 2013).

## 2. Data and methods

Daily assimilated meteorological fields for 1998–2017 over China are obtained from National Centers for Environmental Prediction/National Center for Atmospheric Research (NCEP/NCAR)

Reanalysis 1 provided by the National Oceanic and Atmospheric Administration (NOAA) of the US (Kalney et al., 1996). The dataset has a horizontal resolution of 2.5°×2.5°. Eight meteorological variables are considered here (Table 1), including surface air temperature ($X_1$), relative humidity ($X_2$), precipitation rate ($X_3$), sea-level pressure ($X_4$), pressure tendency ($X_5$), wind speed ($X_6$), and two wind direction indicators ($X_7$, $X_8$). To conduct correlation analysis and PC regression, meteorological data except $X_5$, $X_7$

and $X_8$ are deseasonalized and detrended by subtracting the corresponding centered 30-day moving averages from the original data to focus on day-to-day, synoptic-scale variability. The deseasonalized and detrended data are also normalized to their standard deviations to yield zero means and unit variances.

$PM_{2.5}$ monitoring has been introduced in the national air quality monitoring network in China since 2012 with the published third revision of the "National Ambient Air Quality Standards" (NAAQS)

(Zhang and Cao, 2015). Before that, observational spatial distribution of $PM_{2.5}$ was mostly estimated by satellite retrievals (Ma et al., 2015; van Donkelaar et al., 2010; Xue et al., 2017; Zheng et al., 2016). One of the disadvantages of $PM_{2.5}$ monitoring at present is that there are very few sites with detailed speciation data in China, although short-period studies of $PM_{2.5}$ speciation have been conducted (Cao et al., 2012; Huang et al., 2014; Yang et al., 2005, 2011; Zhang et al., 2014). In this study, hourly mean data of total



PM$_{2.5}$ from 1 Jun 2014 to 30 May 2017 are obtained from the Chinese Ministry of Environmental Protection (MEP). Data are archived from 1497 monitors across China (Fig. 1a), most of which are concentrated in the eastern, northeastern, and southern China, and are made available through one repository website (http://pm25.in). We cross-check and correct the locations of the different monitoring sites, removing unrealistic values and instrumental errors. PM$_{2.5}$ data are then deseasonalized and detrended in the same way as for the meteorological variables.

To conduct the statistical analysis, MEP observations are interpolated using inverse distance weighting onto the same 2.5°×2.5° resolution as that for the NCEP/NCAR data to produce daily mean PM$_{2.5}$ fields for 2014–2017. Sampled values ($z_j$) from sites within a search distance ($d_{max}$) are weighted inversely by their distances ($d_i$) from the cell centroid to produce an average ($z_j$) for each grid cell $j$:

$$z_j = \frac{\sum_{i=1}^{n_j}(1/d_i)^k z_i}{\sum_{i=1}^{n_j}(1/d_i)^k} \tag{1}$$

where $n_j$ is the number of sampled sites for grid cell $j$ and $k$ is the power parameter. We choose $k = 2$ and $d_{max} = 500$ km as recommended by Tai et al. (2010). Figure 1 shows the averaged site and interpolated PM$_{2.5}$ values for 2015 and 2016. As shown in Fig. 1, sites in much of southwestern China (e.g., in the provinces of Tibet and Qinghai) are relatively sparse, leading to likely unrepresentative interpolated values in the corresponding grid cells. These regions are excluded from our analysis.

For the purpose of examining long-term interannual PM$_{2.5}$ variability, we also make use of the annual mean concentrations of surface total PM$_{2.5}$ for 1998–2015 derived from satellite measurements (van Donkelaar et al., 2016). Total column aerosol optical depth (AOD) retrievals from multiple satellite instruments were combined with model simulation based upon comparisons with ground-based sun photometer observations. This combined AOD was related to near-surface PM$_{2.5}$ using the temporally and spatially varying simulated AOD to PM$_{2.5}$ relationship. Resultant annual mean PM$_{2.5}$ values were then calibrated to ground-based PM$_{2.5}$ observations using the Global Geographically Weighted Regression (GWR) method (Brunsdon et al., 1996). Figure S1 shows the spatial variation of the satellite-derived PM$_{2.5}$ over China from van Donkelaar et al. (2016), which has a spatial correlation of $r = 0.70$ with MEP total PM$_{2.5}$ for year 2015.

To project the 2000–2050 effect of climate change on future PM$_{2.5}$, we use meteorological variables in Table 1 archived from an ensemble of 15 climate models participating in the Coupled Model Intercomparison Project Phase 5 (CMIP5) under the representative concentration pathway 8.5 (RCP8.5). We regrid the data from different models into the same 2.5°×2.5° resolution. The details of the models can be found in Table S1.



## 3. Correlations between daily PM$_{2.5}$ and meteorological variables

Here we first discuss the general correlation patterns between PM$_{2.5}$ and individual meteorological variables in China, and highlight what we can and cannot conclude from them. The Pearson's correlation coefficients between each meteorological variable in Table 1 and interpolated daily

total PM$_{2.5}$ are computed for each grid cell from June 2014 to May 2017.

Figure 2 shows the correlation maps for the whole period. Temperature is found to have an overall significant positive correlation with deseasonalized PM$_{2.5}$ in most regions of China (Fig. 2a), with the highest values appearing in BTH and SCB ($r = 0.6$). The correlation map of SLP (Fig. 2d), which is often an indicator of the passage of synoptic systems, has a similar spatial pattern to that with temperature

but with an opposite sign and smaller magnitudes, suggesting that PM$_{2.5}$ tends to be low when SLP is high. The anticorrelation pattern is relatively weaker in southern China. Temperature and SLP are themselves found to be significantly negatively correlated throughout most of China (Fig. S2), and thus it is difficult to conclude whether they are the direct physical drivers of PM$_{2.5}$ variability, or the correlations simply reflect common association with larger meteorological regimes that control PM$_{2.5}$

variability.

Correlation between RH and PM$_{2.5}$ shows different patterns in northern vs. southern China (Fig. 2b). A positive correlation ($r = 0.4$) is seen in BTH, likely reflecting higher PM water content in ambient air which can enhance the uptake of semivolatile components (Dawson et al., 2007b), consistent with previous findings (Wang et al., 2014). In southern China, however, RH is negatively correlated with PM$_{2.5}$,

with larger correlations in SCB and PRD ($r = -0.4$) than in YRD ($r = -0.2$). As can be seen in Fig. 2c, negative correlation of precipitation with PM$_{2.5}$ in southern China is very similar to that of RH in Fig. 2b, likely reflecting the association of high RH with precipitation (Fig. 2c) and onshore wind (Fig. 2f) which can facilitate PM$_{2.5}$ deposition or ventilation (Zhu et al., 2012).

Pressure tendency and wind speed exhibit similar correlation patterns (Fig. 2e-f). Pressure

tendency, another indicator of synoptic-scale motions, is negatively correlated with PM$_{2.5}$ in southern China, including PRD ($r = -0.3$) and in northeastern China, suggesting that PM$_{2.5}$ tends to be low when SLP is increasing. Wind speed is also negatively correlated with PM$_{2.5}$ in similar regions. These patterns are consistent with advecting cold fronts with strong winds helping to ventilate PM$_{2.5}$ in heavily polluted regions (Tai et al., 2012a). Pressure tendency and wind speed have a positive correlation with PM$_{2.5}$ in

northern China and some parts of western China, which may be due to the covarying strong winds and frontal passages promoting the mobilization of mineral dust from the semiarid regions and deserts there.

Figure 2g shows the correlation of wind direction with PM$_{2.5}$, in which arrow directions indicate wind directions associated with increasing PM$_{2.5}$. For instance, PM$_{2.5}$ increases with southeasterly wind for the whole eastern and northeastern China with a correlation of $r = 0.3$ on average. This relationship

suggests that northwesterly wind tends to ventilate PM$_{2.5}$ in most of China. Two divergent wind patterns



are seen, one in central China and one in Teklimakan desert, and their positions mirror regions with the highest $PM_{2.5}$ concentrations in Fig 1b. This result implies that wind transports pollutants from source regions to the peripheries.

A generally consistent correlation among neighbouring grid cells may be associated with
synoptic effects because the correlation pattern extends to a synoptic regional length scale. The correlation maps for most of the meteorological variables in Fig. 2 show such an effect. The commonality among the correlation patterns of $PM_{2.5}$ with different meteorological variables, which among themselves have various degrees of correlation, renders the interpretation of individual $PM_{2.5}$-meteorology relationships more difficult because the true driver of $PM_{2.5}$ variability may be masked by the collinearity among
meteorological variables (as is pointed out above for the case of temperature and SLP). Whenever a strong correlation between $PM_{2.5}$ and a given local meteorological variable (e.g., temperature, RH, precipitation, wind speed) is found, there can be three interpretations: (1) this variable is truly the physical driver for $PM_{2.5}$ variability; (2) at least part of the correlation may arise from the correlation of this variable with another local variable that is the true physical driver; and (3) at least part of the correlation may reflect
common association with a larger, synoptic-scale phenomenon that drives $PM_{2.5}$ variability. To quantitatively differentiate between these possibilities and to ascertain the roles of local meteorology vs. synoptic-scale circulation on $PM_{2.5}$ variability, we conduct a principal component analysis (PCA) on the eight meteorological variables to capture their common covariations in an ensemble of independent meteorological modes. We follow Tai et al. (2012a), and regress daily $PM_{2.5}$ on the resulting principal
component (PC) time series to identify the dominant synoptic drivers of $PM_{2.5}$ variability.

## 4. Dominant meteorological modes for daily $PM_{2.5}$ variability based on principal component regression

We perform PCA on the eight meteorological variables for 1998–2017 in Table 1, focusing on the four major metropolitan regions in China (BTH, YRD, PRD and SCB). We use this longer period of
meteorological data for the PCA despite the relatively short time history of $PM_{2.5}$ data from MEP (2014–2017) because we aim to characterize the climatologically important synoptic systems in China. The longer period also overlaps with the annual mean $PM_{2.5}$ data available for quantifying interannual variability (see Sect. 5), and so a unified set of meteorological modes can be used to explain both daily and interannual $PM_{2.5}$ variability. We conduct PCA for individual seasons and for the whole period. All
gridded daily meteorological data are spatially averaged over the grid cells covering each of the four regions, deseasonalized, and normalized to yield zero means and unit variances, as described above. The resulting time series for each region are then decomposed to produce the PC time series ($U_j = U_1, …, U_8$):

$$U_j(t) = \sum_{k=1}^{8} \alpha_{kj} \frac{X_k(t) - \bar{X}_k}{s_k} \tag{2}$$



where $X_k$ represents the regionally averaged meteorological fields in Table 1, $\bar{X}$ and $s_k$ are the temporal mean and standard deviation of $X_k$, and $\alpha_{kj}$ are the elements of the transformation matrix (i.e., eigenvector or empirical orthogonal function, EOF) of PCA. The PC time series are ranked by their variances $\lambda$, with the leading three to four PCs capturing most of the meteorological variability (Wilks, 2011). For example,

the first four PCs for the BTH region explain 76% of total meteorological variability. The last few PCs with variances $\lambda < 1$ are truncated using the Kaiser's rule since they likely represent noises (Wilks, 2011). Each PC represents a distinct meteorological mode, the physical meaning of which is reflected by the values of $\alpha_{kj}$ in Eq. (2) and verified by cross-examination of synoptic weather maps.

For each region, we then extract the PCs for 2014–2017 only, and construct a principal

component regression (PCR) model for deseasonalized, regionally averaged daily $PM_{2.5}$ ($y$, µg m$^{-3}$) on the daily PC values ($U_j$) for 2014–2017, both for the whole period and for individual seasons:

$$y(t) = \sum_{j=1}^{N} \beta_j U_j(t) \tag{3}$$

where $\beta_j$ is the regression coefficient (µg m$^{-3}$), and $N$ the number of PCs retained after truncation (mostly 3 to 4).

We define a dominant meteorological mode seasonally or annually by computing the ratio of the resulting regression sum of squares ($SSR_j$) to total sum of squares (SST) for each PC:

$$\frac{SSR_j}{SST} = \frac{\sum_t [\beta_j U_j(t)]^2}{\sum_t \{[y(t) - \bar{y}]/s_y\}^2} \tag{4}$$

This ratio characterizes the fraction of variance of daily $PM_{2.5}$ that can be explained by the $j^{th}$ PC in the PCR model. The PC having the largest SSR/SST is deemed the dominant meteorological mode for that

region. Any PC which has an SSR/SST more than half of that of the dominant PC in a given season is also recognized as an important PC for that region.

Here we discuss the synoptic meteorological systems that dominate $PM_{2.5}$ variability on annual timescales for each region. Discussion of regimes that control $PM_{2.5}$ on seasonal timescales, as well as information on the values of SSR/SST and $\beta$, is included in the supplementary materials.

Figure 3 shows the dominant meteorological mode in BTH, which explains nearly 36% of $PM_{2.5}$ variability throughout the year. Figure 3a shows a strong anticorrelation between the time series of this mode and deseasonalized observed total $PM_{2.5}$ for the sample month of December 2014. Figure 3b shows the meteorological composition of the EOF of this annually dominant mode, with a positive phase consisting of low temperature, high SLP, and strong northwesterly winds. The error bars represent two

standard errors of the meteorological composition, computed by the formula shown in Sect. S1. Similar loadings are seen for winter, spring, and fall. We choose 30 Dec 2014 as a representative day with PC changing from negative to positive phase to explain the physical meaning of this PC. As seen in the weather map (Fig. 3c), the positive phase of PC1 represents a high-pressure system associated with the Siberian high with dry cold fronts sweeping across BTH from northwest to southeast. The Siberian high

is the driver of the winter monsoon in East Asia, and such northwesterly flow efficiently advects $PM_{2.5}$



across BTH. Figure 3c shows a strongly decreasing temperature gradient and increasing pressure tendency originating from the Siberian high. $PM_{2.5}$ concentration decreases by nearly 240 μg m$^{-3}$ over 29 to 31 Dec (Fig. 3a). In addition to cold fronts from the Siberian high, easterly onshore flow with high humidity and southerly monsoon also control daily $PM_{2.5}$ variability in spring and summer, with 18% and 17%

variability of $PM_{2.5}$ explained, respectively (see Sect. S2).

Figure 4 shows the dominant mode in YRD. This mode is important in spring, fall and winter, and contributes up to 14% of the $PM_{2.5}$ variability for the whole year. The two time series of PC1 and $PM_{2.5}$ demonstrate anticorrelation with each other in March 2014 (Fig. 4a). The positive phase of this mode consists of low temperature, high RH and rainfall, high and decreasing pressure, and strong easterly

winds (Fig. 4b). This set of meteorological phenomena is characteristic of onshore flow with rainfall, as demonstrated by the weather map on 25 Mar 2015, which shows cold and moist easterly winds originated from the high pressure centered over the East Sea. Such winds sweep away pollutants and decrease $PM_{2.5}$ concentration by 30 μg m$^{-3}$ (Fig. 4c), and the associated rainfall also wash out $PM_{2.5}$. The negative phase of this mode, as represented on 18 Mar 2015, shows anticyclonic flow leading to accumulation of $PM_{2.5}$

(Fig. 4d). In addition to onshore flow, PCA for summer alone indicates that summertime low-pressure systems also deplete $PM_{2.5}$, likely due to the associated precipitation, explaining 24% $PM_{2.5}$ variability. This PC is also sometimes characterized by northward-propagating tropical cyclones, with strong wind and rainfall (see Sect. S3).

Figure 5 shows the dominant mode for explaining $PM_{2.5}$ variability in PRD. This mode is

dominant in spring, fall and winter, and in total contributes to 22% variability of $PM_{2.5}$ throughout the year. Fig. 5a reveals a negative correlation between the PC for this mode and $PM_{2.5}$ in October 2014. The positive phase of this mode consists of high RH, precipitation, increasing pressure and strong northerly winds (Fig. 5b). This set of meteorological phenomena represents a cold-frontal rainstorm, as demonstrated by the weather map in Fig. 5c, which shows a frontal rain belt coinciding with the positive

phase of PC1 on 21 Oct 2014. Pressure contours were advected southward by northerly winds, and a regional rain belt brought maximum rainfall of up to 15 mm d$^{-1}$ to southern China. In general for this mode, advancing cold air sweeps from north to south and lifts the warmer and moister air, leading to precipitation and sometimes thunderstorms. In addition to cold-frontal rainstorms, summertime PCA also shows that the air quality in summer PRD is also influenced by rainfall from low-pressure troughs as well

as by landfalls of tropical cyclones (see Fig. S10 & S11). These two modes explain 18% and 15% of summertime $PM_{2.5}$ variability, respectively. The troughs cause rainfall that scavenges pollutants; tropical cyclones having landfalls to the east of PRD cause inversion layers that trap pollutants and degradate air quality (see Sect. S4).

Figure 6 shows the dominant mode in SCB in winter, which has a negative correlation with

$PM_{2.5}$, as shown for the sample month of January 2015 (Fig. 6a). This mode dominates $PM_{2.5}$ variability all year round, explaining 25% of its day-to-day variability. PCA shows that its positive phase is





characterized by low temperature, high SLP and weak northwesterly winds (Fig. 6b), which resembles the dominant EOF in BTH. This mode is characterized by a northwesterly flow also associated with the Siberian high. On 29 Jan 2015, the Siberian High was situated southeast to Lake Baikal (Fig. 6c), advecting a clean, northwesterly cold front toward SCB and ventilating $PM_{2.5}$ by 150 μg m$^{-3}$ over 25 to

29 Jan. On 24 January, this mode was in its negative phase and SCB was under a relatively mild weather (Fig. 6d), while $PM_{2.5}$ was at a local maximum (Fig. 6a). In addition to cold-frontal passages, rainfall also drives $PM_{2.5}$ variability especially in winter and spring, explaining 18% and 16% of wintertime and springtime $PM_{2.5}$ variability, respectively. This mode represents a cold-frontal rain system that promotes wet deposition of pollutants (see Sect. S5).

**5. Synoptic frequency as a metric for climate change impact on $PM_{2.5}$**

Future climate change can significantly affect synoptic-scale circulation patterns and local meteorology, modifying the transport and deposition of $PM_{2.5}$ (Fiore et al., 2015; Jiang et al., 2013; Mickley et al., 2004). Based on the demonstrated strong relationships of synoptic circulation and local meteorology on daily $PM_{2.5}$, we build a regression model to infer how interannual variations of local and

synoptic meteorology affect interannual $PM_{2.5}$ variability, which we then apply to future climate projections. This approach allows us to evaluate the potential impacts of climate change on $PM_{2.5}$ air quality. Here we adopt the PCA-spectral analysis approach, namely, to apply a Fast Fourier Transform (FFT) to the daily time series of the dominant PCs for all seasons to extract the median frequencies from the resulting spectra. We use the same PCs generated from 1998–2017 NCEP/NCAR meteorological data

(Sect. 4), and smooth the resulting FFT spectra with a second-order autoregressive filter (Wilks, 2011). We focus on BTH as a case study. For example, spectral analysis shows that the Siberian high fluctuates between 58 and 67 times per year on average, and has a climatological frequency of 63 yr$^{-1}$ averaged over 1998–2015.

Satellite-retrieved $PM_{2.5}$ has large uncertainties in seasonal mean values, and thus we make use

of only the annual mean $PM_{2.5}$ values for building our regression model. We construct a multiple linear regression (MLR) model for the 1998–2015 satellite-retrieved annual mean $PM_{2.5}$ over BTH by spatially averaging the gridboxes covering the region. In selecting predictor variables, we consider the annual mean local meteorological variables in Table 1 (except SLP tendency ($X_5$) and the two wind direction indicators ($X_7$, $X_8$), whose averages are often nearly zero), as well as the annual median frequencies of synoptic

circulation patterns from all individual seasons diagnosed from spectral analysis. The predictand (annual mean $PM_{2.5}$) and potential predictors are detrended by subtracting from them the respective 7-year moving averages in order to remove long-term trends driven by emission changes. We adopt a forward selection approach (Wilks, 2011) to identify which climatic variables explain the greatest amount of interannual $PM_{2.5}$ variability, starting from the one explaining the largest percentage of $PM_{2.5}$ variability (having the





largest adjusted $R^2$ value), and adding predictor variables until the enhancement in adjusted $R^2$ given by an additional predictor is less than 0.05. Variables that lead to a large variance inflation factor (>2) are also excluded to avoid the issue of multicollinearity. Typically the forward selection algorithm does not yield more than three predictor variables for interannual $PM_{2.5}$ variability.

5        Table 2 shows the interannual $PM_{2.5}$ variability explained by the predictors, the corresponding regression coefficients and the $p$-values for the BTH region. The two predictors selected by the forward selection algorithm are the frequency of the first PC in spring (i.e., the springtime Siberian high, Figure S4) and annual mean RH. Figure 7 shows the correlation of detrended annual mean $PM_{2.5}$ with detrended annual mean RH and the frequency of fluctuation of the springtime Siberian high. The negative

correlation ($r = -0.51$) between springtime PC frequency and annual $PM_{2.5}$ indicates that more frequent occurrences of high-pressure systems further north especially during spring help ventilate $PM_{2.5}$ in BTH and influence annual mean $PM_{2.5}$ here. This is consistent with the relationship we found between $PM_{2.5}$ and Siberian high on the daily timescale (Sect. 4). Annual mean RH has a positive correlation with $PM_{2.5}$ ($r = 0.49$), which is consistent with Sect. 3 where we found higher RH coinciding with higher $PM_{2.5}$ on

the daily timescale. Adding RH helps explain an additional 9% interannual $PM_{2.5}$ variability, and the two predictors in total give an adjusted $R^2$ value of 31%.

       Our findings show that meteorological effects on daily $PM_{2.5}$ at least in part contribute to interannual variability $PM_{2.5}$, a finding which we can exploit to estimate future changes in $PM_{2.5}$. To this end, we extract the meteorological variables in Table 1 from the results from 15 models in the Climate

Model Intercomparison Project Phase 5 (CMIP5) for 1996–2005 and 2046–2055 under the RCP8.5 scenario (Table S1). This scenario represents a business-as-usual future. We diagnose the 2000–2050 changes in the decadal averages of these variables and the median frequencies of the constructed PCs, which we then combine with the regression coefficients in Table 2. This step yields an estimate for 2000–2050 change in annual mean $PM_{2.5}$ due to climate change alone. We use a Monte-Carlo approach to

characterize the probability distribution and statistical significance of the changes in $PM_{2.5}$ concentration arising from the uncertainties of the regression coefficients in the MLR model, as well as from the differences in model physics among CMIP5 models. Our approach involves drawing regression coefficients from the MLR model assuming a Gaussian distribution, and $PM_{2.5}$ changes from the 15 CMIP5 models assuming a uniform distribution.

Figure 8 shows the future changes of $PM_{2.5}$ concentrations with the corresponding changes in future meteorology. Changes in RH among CMIP5 models show high inconsistency, with values ranging from –2% to +3% (Figure 8a). The ensemble mean of CMIP5 models shows a statistically insignificant increase ($p$-value = 0.32) of RH of 0.2±1.78 percentage point by 2050 in BTH, consistent with a future prediction of an increase by < 1% over BTH in IPCC AR5 (Fig. 12.21 in Collins et al., 2013). Past

modeling studies show that RH remains nearly constant on climatological timescales and continental spatial scales (Randall et al., 2007), while recent investigation shows that near-surface RH decreases over



most land areas globally (O'Gorman and Muller, 2010). IPCC AR5 (2013) shows that the regional mean RH in BTH changes by less than one standard deviation of interannual variability by year 2065, and the variability is dominated more by naturally occurring processes than by human activities.

We find an overall *likely* (i.e., 66–100% likelihood according to Box TS.1 in Stocker et al., 2013), statistically significant increase ($p$-value = 0.046) in the frequency of synoptic-scale fluctuation of the Siberian high by $1.1\pm2.01$ yr$^{-1}$ by the 2050s (Figure 8a). Only three of the 15 models project a decrease in this synoptic frequency. The generally increasing frequency is possibly driven by the future reduction in meridional temperature gradient, which decreases the intensity of the midlatitude jets and favors the amplification and persistence of surface anticyclones (e.g., Francis and Vavrus, 2012; Zhang et al., 2012).

Francis and Vavrus (2012) showed that the upper tropospheric midlatitude jet (in the form of Rossby wave) exhibited reduced zonal velocity and augmented wave amplitude under warming over 1979–2010, which may have led to an increase in atmospheric blocking events (Barriopedro et al., 2006) and an enhancement in the likelihood of cold surges from the Siberian high. In another multi-model study, Park et al. (2011), however, found no significant correlation between cold surge occurrences and surface air

temperature over East Asia, and thereby concluded that cold surge occurrences would remain constant in frequency under a warming climate. Our results based on PCA-spectral analysis show a modest increase instead of unchanging frequency in synoptic-scale fluctuation of the Siberian High in the future.

Figure 8b shows the corresponding PM$_{2.5}$ changes. Averaged across model results, we find that PM$_{2.5}$ will increase $+0.21\pm1.78$ µg m$^{-3}$ ($p$-value = 0.67) due to changing RH, but decrease by $-0.34\pm0.63$

µg m$^{-3}$ ($p$-value = 0.046) due to increasing frequency of cold fronts. We show that under climate change, changes in RH and the frequency of fluctuation in the Siberian High would constitute a climate "penalty" and "benefit" for PM$_{2.5}$ air quality, respectively. These two effects largely offset each other, resulting in a combined mean PM$_{2.5}$ change by only about $-0.13$ µg m$^{-3}$ and representing a slight overall climate benefit. The Monta-Carlo simulation shows that the standard error of local-meteorology- and synoptic-

frequency-induced PM$_{2.5}$ changes are 1.97 µg m$^{-3}$ and 0.71 µg m$^{-3}$ respectively (Fig. 8c), much larger than the mean values of PM$_{2.5}$ changes. We find that most of the uncertainty stems from large intermodel differences in the future projections of RH and, to a lesser extent, in those of synoptic frequency in CMIP5. The regression coefficients have relatively moderate standard errors (Table 2).

## 6. Conclusions and discussion

In this study we use a combination of multivariate statistical methods to investigate the local and synoptic meteorological effects on daily and interannual variability of PM$_{2.5}$ in China. Based on the resulting statistical relationships between PM$_{2.5}$ with annual mean meteorological variables and synoptic frequencies, we also project future PM$_{2.5}$ changes in the Beijing-Tianjin-Hebei (BTH) region. First, we find strong correlations between daily observed PM$_{2.5}$ and individual meteorological variables in China



over 2014–2017, and the spatial patterns of correlations suggest common association of these variables with synoptic circulation and transport. We therefore apply PCA on spatially averaged meteorological variables for four major metropolitan regions (BTH, YRD, PRD, SCB) for 1998–2017 (for all seasons and for the whole period) to diagnose the dominant synoptic meteorological modes, and the time series

of these modes are used as predictor variables in a MLR model to explain day-to-day $PM_{2.5}$ variability for each region. We find that, in BTH, the presence of the Siberian high strongly controls $PM_{2.5}$ levels. Northerly monsoonal flows and advecting cold fronts from the Siberian high play key roles in ventilating $PM_{2.5}$ in BTH for all seasons except JJA. In YRD, onshore wind with precipitation from the East Sea is the dominant meteorological mode, effectively scavenging $PM_{2.5}$ for all seasons except JJA. In PRD,

frontal rain is a key driver reducing $PM_{2.5}$ by wet deposition for all seasons except JJA. In SCB, the Siberian high plays a key role in bringing clean air from the north that effectively dilutes pollution for all seasons. Different synoptic meteorological regimes in different seasons explain about 16–37% of $PM_{2.5}$ variability in 2014–2017.

We further show that the long-term fluctuations in the frequencies of the dominant synoptic

modes also shape interannual variability of $PM_{2.5}$. Using the BTH region as a showcase, we use regionally averaged annual mean local meteorological variables and annual median frequencies of the dominant synoptic modes of all individual seasons as potential predictors in a forward-selection MLR model to explain the interannual variability of satellite-derived annual mean $PM_{2.5}$ over 1998–2015. The forward selection model finds two significant predictors, namely, the frequency of springtime frontal passages

(which indicates the interannual fluctuation in the strength of the Siberian high) and annual mean RH, with observed $PM_{2.5}$-to-climate sensitivities of $-0.31\pm0.16$ µg m$^{-3}$ yr and $1.00\pm0.57$ µg m$^{-3}$ %$^{-1}$, which together explain 31% of the variability of annual mean $PM_{2.5}$. The signs of correlations between $PM_{2.5}$ and the two predictors are also consistent with that from the daily PC regression analysis, showing a broad consistency in $PM_{2.5}$-meteorology relationships across different timescales.

We further address the effect of 1996–2055 climate change on future $PM_{2.5}$ air quality, using an ensemble of 15 CMIP5 climate model outputs under the RCP8.5 scenario. Twelve out of 15 models show an increase in the frequency of strength fluctuation of the Siberian high by 1.1 yr$^{-1}$ on average. Nine out of 15 models show a modest increase in future RH by 0.2% in average. Intermodel differences in the projected changes in RH are much larger than that in synoptic frequency of fluctuation in the Siberan

High, owing to the high inconsistency in future projections of atmospheric humidity, especially on a regional scale (IPCC, 2013). Combining the ensemble projections of RH and synoptic frequency with the $PM_{2.5}$-to-climate sensitivities from our statistical model, we project by 2050s a *more likely than not* (~60% likelihood) increase in $PM_{2.5}$ of $0.21\pm2.00$ µg m$^{-3}$ due to RH, and a *likely* (~80% likelihood) decrease in $PM_{2.5}$ of $-0.34\pm0.69$ µg m$^{-3}$ due to increasing frequency in the fluctuation of the Siberian High. The

resulting combined effect on $PM_{2.5}$ is a *more likely than not* (>50% likelihood) decrease of $-0.13\pm2.10$ µg m$^{-3}$. Our prediction is comparable in magnitude with other studies (e.g., Jiang et al., 2013), as well as




future predictions done for the US (Shen et al., 2017; Tai et al., 2012b; Pye et al., 2009; Avise et al., 2009) and Europe (Juda-Rezler et al., 2012). Jiang et al. (2013) projected changes of $PM_{2.5}$ over China due to climate change alone under IPCC A1B scenario, and the resulting change over BTH is about $+1\ \mu g\ m^{-3}$ averaged annually. They attributed their predictions to: 1) changing precipitation that leads to a change

in wet deposition; and 2) increasing temperature that results in more volatilization of nitrate and ammonium, which differs from our conclusion that RH and cold fronts dominate the total $PM_{2.5}$ response. Our statistical results (for BTH only) do not see significant relationships between temperature and $PM_{2.5}$ ($r = 0.18$) nor between rainfall and $PM_{2.5}$ ($r = 0.20$) on an interannual timescale, despite strong correlations on a daily timescale. This discrepancy between empirical results and process-based model results may

stem from the inadequacy of satellite-derived $PM_{2.5}$ in capturing the variability caused by volatilization effect, an inadequate process-based model representation of the $PM_{2.5}$-temperature relationship (Shen et al., 2017), and uncertainties in emissions of PM precursors in the process-based model.

There are two major limitations of the statistical approach developed in this study. First, due to accuracy constraints of the satellite-derived $PM_{2.5}$ concentrations, we could only use annual mean instead

of seasonal mean $PM_{2.5}$ as the basis for interannual regression and future projections. Shen et al. (2017) showed that $PM_{2.5}$ responds to meteorological conditions differently in different seasons in the US. Due to the short period of surface monitoring data (see Sect. 2), we rely on the annual mean satellite-derived $PM_{2.5}$ with no seasonality in this study, and thus no seasonal predictions of $PM_{2.5}$ are possible. Another limitation is that the statistical projections rely on the assumption that the $PM_{2.5}$-to-climate sensitivities

will be more or less constant in the future. This assumption may be acceptable for near-future projections (Fiore et al., 2012; IPCC, 2013), but is more vulnerable for multidecadal projections especially as significant changes in emission levels may alter the chemical nature of total $PM_{2.5}$ and thus the interactions with meteorology. While the process-based modeling studies of the future evolution of $PM_{2.5}$-meteorology relationships under varying levels of emissions in China are much warranted, the empirical

relationships as diagnosed from investigation of historical data in this study are valuable in providing a basis for testing and validating the process-based model sensitivities of $PM_{2.5}$ air quality to climate change.

**Acknowledgement**

This work was supported by a faculty start-up allowance from the Croucher Foundation and The Chinese

University of Hong Kong (CUHK) (project ID: 6903601, 4930041) given to the principal investigator, Amos P. K. Tai, as well as a Vice-Chancellor Discretionary Fund (Project ID: 4930744) from CUHK given to the Institute of Environment, Energy and Sustainability.





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



**Table 1.** Meteorological variables considered in this study[a].

| Variable | Meteorological parameter (abbreviation, unit) |
|---|---|
| $X_1$ | Surface air temperature ($T$ or SAT, K)[b] |
| $X_2$ | Surface air relative humidity (RH, %)[b] |
| $X_3$ | Surface precipitation rate (Prec, mm d$^{-1}$)[b] |
| $X_4$ | Sea level pressure (SLP, hPa) |
| $X_5$ | Sea level pressure tendency (d$P$/dt, hPa d$^{-1}$) |
| $X_6$ | Surface wind speed (Wind, m s$^{-1}$)[b, c] |
| $X_7$ | West-east direction indicator (cos$\theta$, dimensionless) |
| $X_8$ | South-north direction indicator (sin$\theta$, dimensionless) |

5   [a] From the National Center for Environmental Prediction/National Center for Atmospheric Research (NCEP/NCAR) Reanalysis 1 for 1998–2017. All data are 24-h averages and are deseasonalized as described in the text.
[b] Surface data are from 0.995 sigma level.
[c] Calculated from the horizontal wind vectors ($u$, $v$).
10   [d] $\theta$ is the angle of the horizontal wind vector counterclockwise from the east. Positive values of $X_7$ and $X_8$ indicate westerly and southerly winds, respectively.



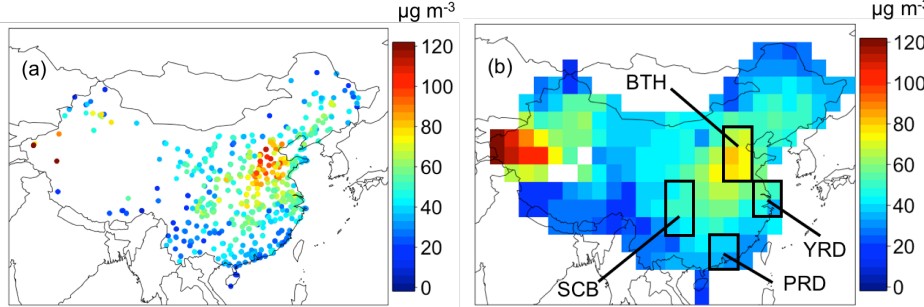

**Fig. 1.** Average (a) site and (b) gridded 2.5°×2.5° total PM$_{2.5}$ concentrations (µg m$^{-3}$) of China during
the years 2015–2016 obtained from the Chinese Ministry of Environmental Protection (MEP,
http://pm25.in). Gridded data are obtained by spatially interpolating site data using an inverse weighting
5   method as in Tai et al. (2010). The four main regions of our study are indicated in panel (b): the
Beijing-Tianjin-Hebei (BTH), the Yangtze River Delta (YRD), the Pearl River Delta (PRD), and the
Sichuan Basin (SCB).




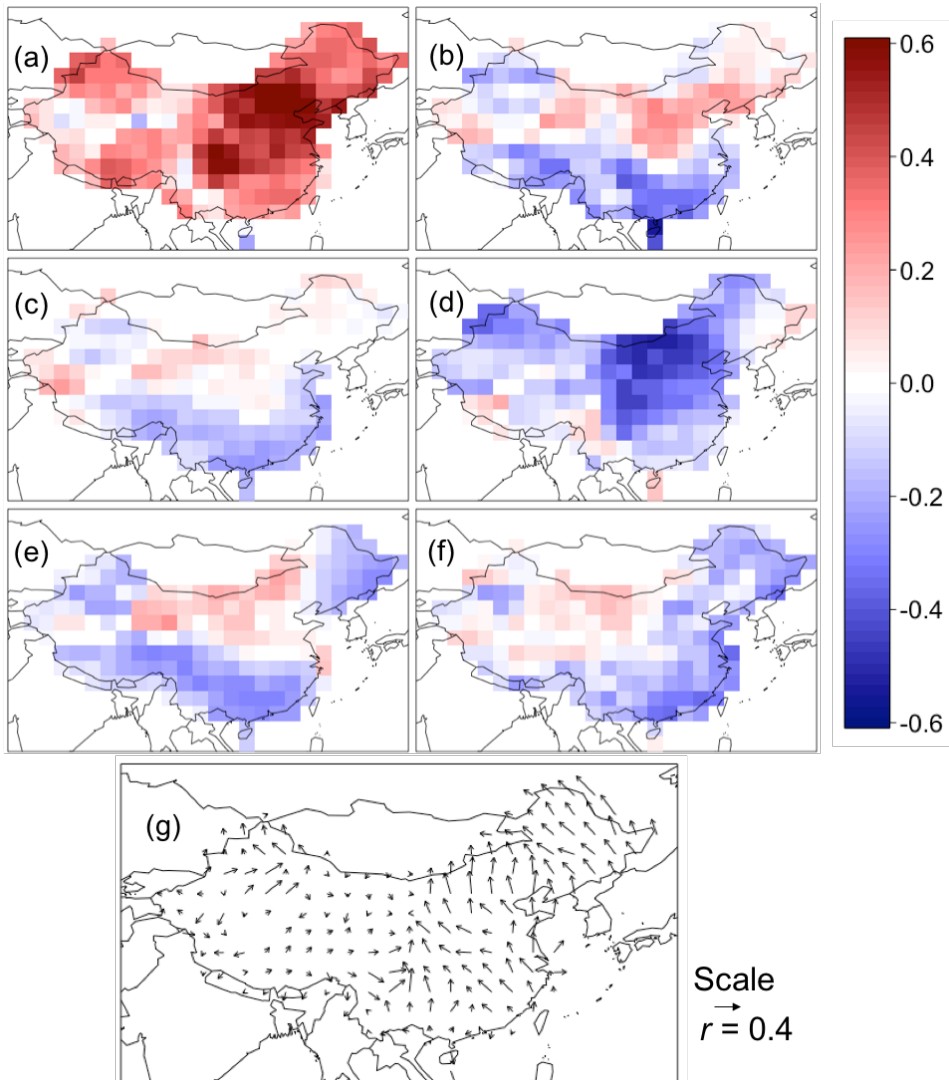

**Fig. 2.** Correlation coefficients of daily $PM_{2.5}$ with different meteorological variables in Table 1, including (a) surface air temperature ($X_1$, K), (b) relative humidity ($X_2$, %), (c) precipitation ($X_3$, mm d$^{-1}$), (d) sea level pressure ($X_4$, hPa), (e) pressure tendency ($X_5$, hPa d$^{-1}$), (f) wind speed ($X_6$, m s$^{-1}$), and (g) wind direction ($X_7$ and $X_8$, unitless), for China from Jun 2014 to May 2017. $PM_{2.5}$ data are from MEP. Meteorological data are deseasonalized by subtracting 30-day moving averages and normalized, and daily total $PM_{2.5}$ are also deseasonalized the same way to focus on day-to-day variability. Only values with significant correlations at $p$-value $\leq 0.05$ are shown, except in panel (g), which shows correlations with wind direction regardless of significance.





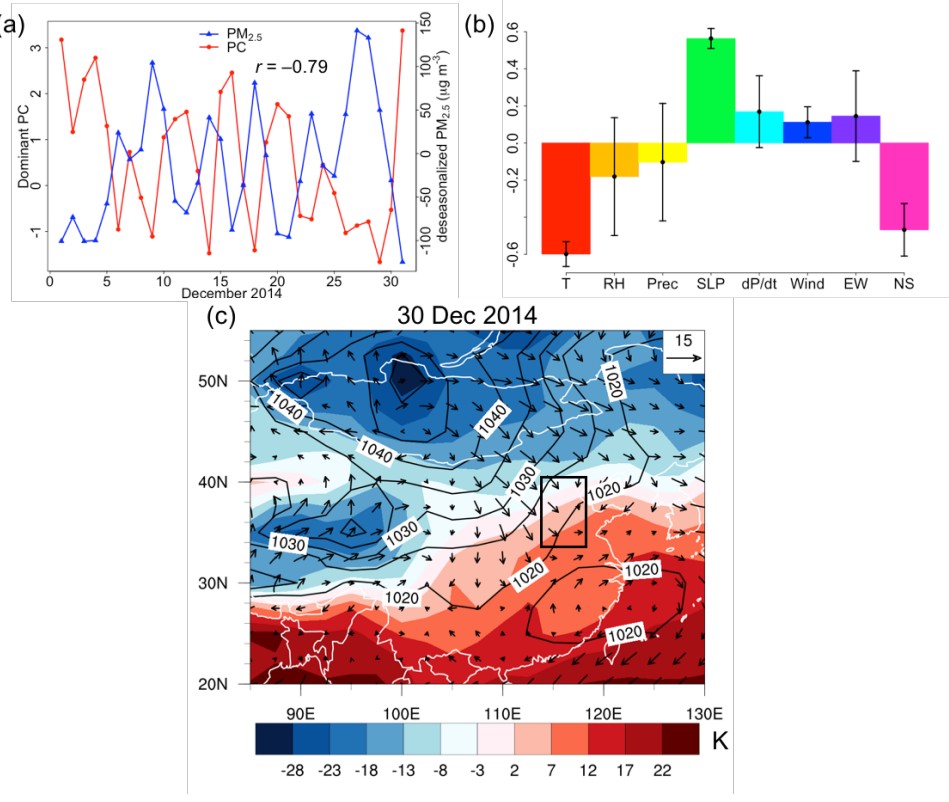

**Fig. 3.** Annually dominant meteorological mode for observed $PM_{2.5}$ variability in the Beijing-Tianjin-Hebei (BTH). (a) Timeseries of deseasonalized observed total $PM_{2.5}$ concentrations and the principal component (PC) time series in the sample month of December 2014. (b) Composition of this mode as determined by the coefficients $\alpha_{kj}$, with error bars showing two standard deviations of the eigenvector coefficients. Meteorological variables are listed in Table 1. (c) Synoptic weather map on 30 Dec 2014 with temperature (K) as shaded colors, wind speed (m s$^{-1}$) as vectors and sea level pressure (hPa) as contours. The rectangle indicates BTH. The weather map, which shows an example of positive influence of the mode, is plotted using NCEP/NCAR reanalysis I data.





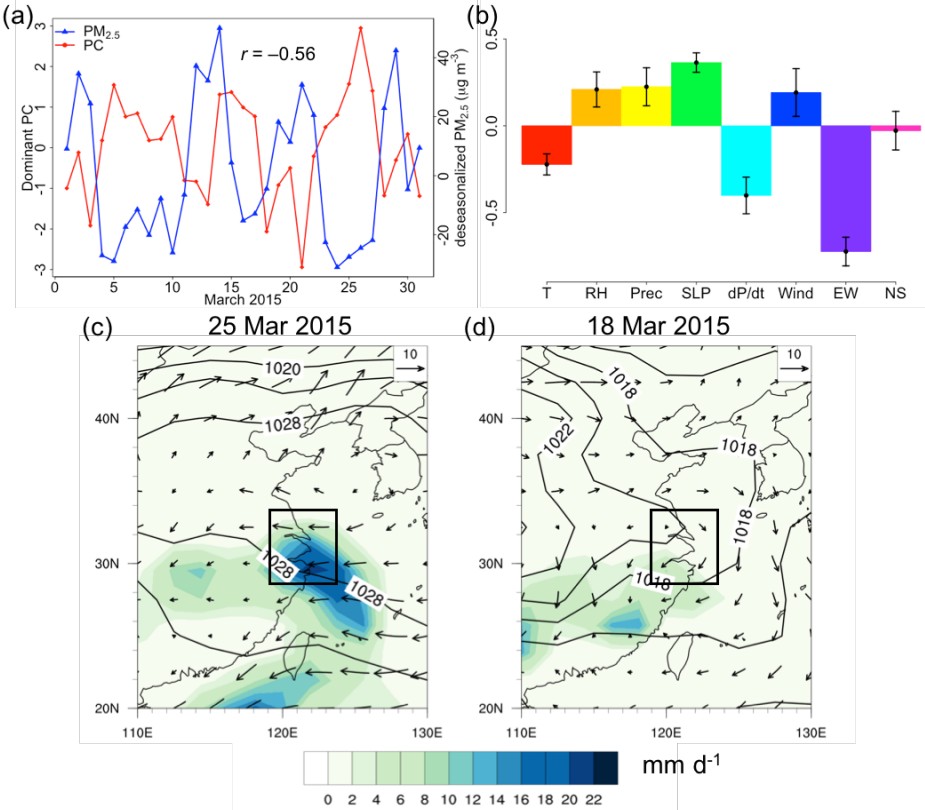

**Fig. 4.** Same as Fig. 3 but for the Yangtze River Delta (YRD). (a) Deseasonalized total PM$_{2.5}$ concentrations and the PC time series in the sample month of March 2015. (b) Composition of this dominant mode as determined by the coefficients $\alpha_{kj}$. (c-d) Synoptic weather charts on 25 and 18 Mar 2015, with temperature (K) shown as shaded colors, wind speed (m s$^{-1}$) as vectors and sea level pressure (hPa) as contours. Panel (c) shows the positive influence characterized by onshore wind with rainfall that corresponds to decreasing PM$_{2.5}$, while panel (d) shows the negative influence with little wind on YRD. The rectangles indicate YRD.

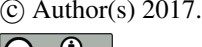



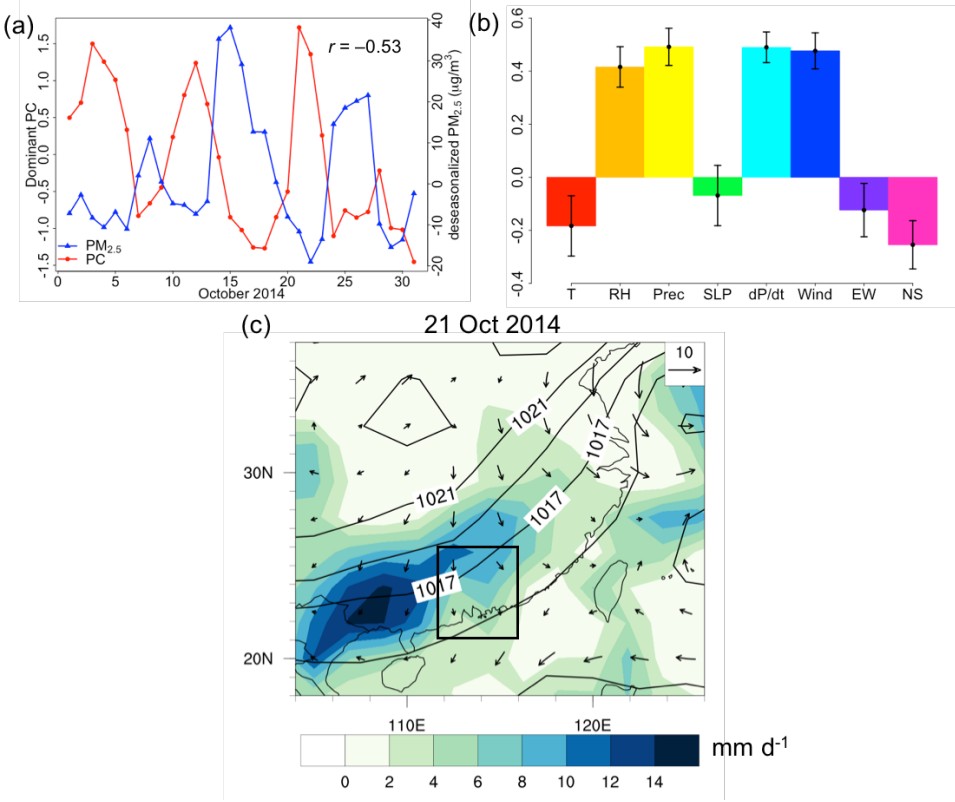

**Fig. 5.** Same as Fig. 3 but for fall in the Pearl River Delta (PRD). (a) Deseasonalized total PM$_{2.5}$ concentrations and the PC time series in the sample month of October 2014. (b) Composition of this dominant mode as measured by the coefficients $\alpha_{kj}$. (c) Synoptic weather map on 21 Oct 2014,
5   corresponding to the positive influence from the mode, with precipitation (mm d$^{-1}$) as shaded colors, wind speed (m s$^{-1}$) as vectors and sea level pressure (hPa) as contours. The rectangle indicates PRD.



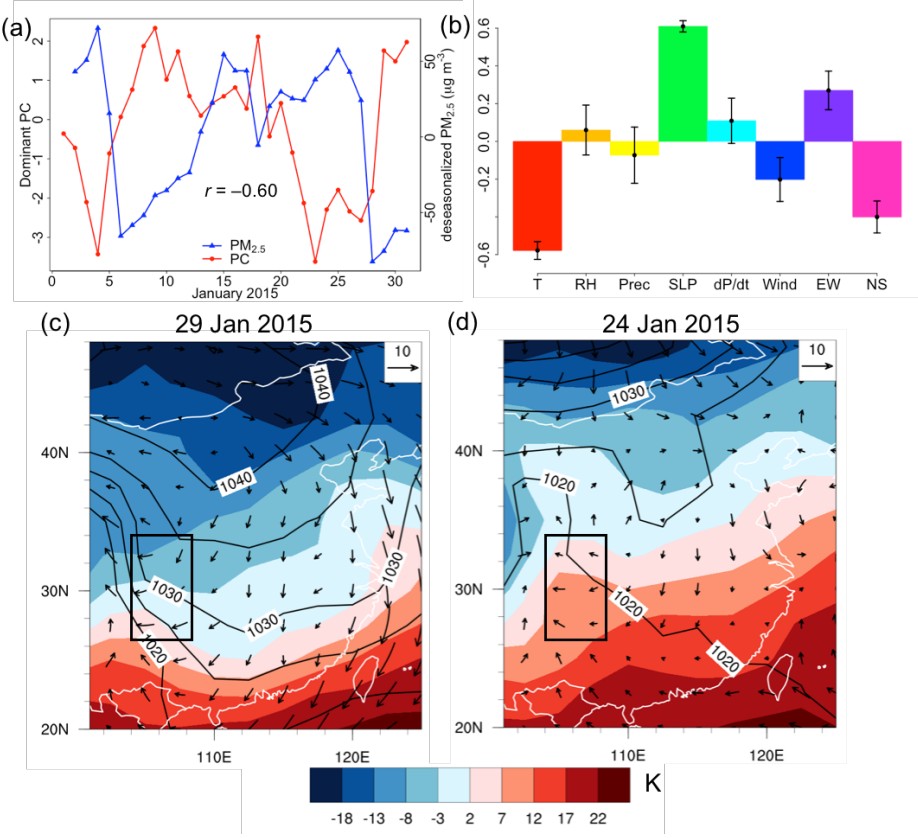

**Fig. 6.** Same as Fig. 3 but for winter in the Sichuan Basin (SCB). (a) Deseasonalized total PM$_{2.5}$ concentrations and the PC time series in the sample month of January 2015. (b) Composition of this dominant mode as measured by the coefficients $\alpha_{kj}$. (c-d) Synoptic weather maps on 29 and 24 Jan 2015. Panel (c) shows the positive influence characterized by a cold front from the Siberian high that advects PM$_{2.5}$ away, while panel (d) shows the negative influence characterized by stagnation over SCB. Temperature (K) is shown as shaded colors, wind speed (m s$^{-1}$) as vectors and sea level pressure (hPa) as contours. The rectangles indicate SCB.





**Table 2.** Regression model that explains interannual $PM_{2.5}$ variability in Beijing-Tianjin-Hebei (BTH).

| | Frequency of springtime Siberian High | Relative humidity |
|---|---|---|
| $PM_{2.5}$ sensitivity | $-0.31$ µg m$^{-3}$ yr | $1.00$ µg m$^{-3}$ %$^{-1}$ |
| Standard error | $\pm0.16$ µg m$^{-3}$ yr | $\pm0.57$ µg m$^{-3}$ %$^{-1}$ |
| $p$-value for each predictor | 0.0776 | 0.0977 |
| Adjusted $R^2$ value | 0.309 | |
| F-statistic | 4.81 | |
| Total $p$-value | 0.0244 | |





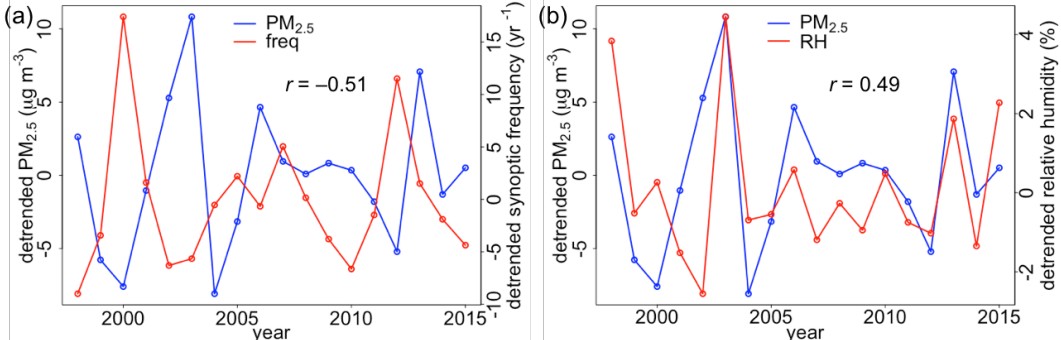

5    **Fig. 7.** Detrended annual mean total $PM_{2.5}$ concentration and climate variables chosen by the forward
selection model from 1998–2015, including (a) annual mean frequency of springtime Siberian High ($r =$
$-0.51$) and (b) relative humidity ($r = 0.49$). Annual mean surface $PM_{2.5}$ concentrations are derived from
satellite aerosol optical depth by van Donkelaar et al. (2016). All variables are detrended by subtracting
the 7-year moving averages from the annual mean values.



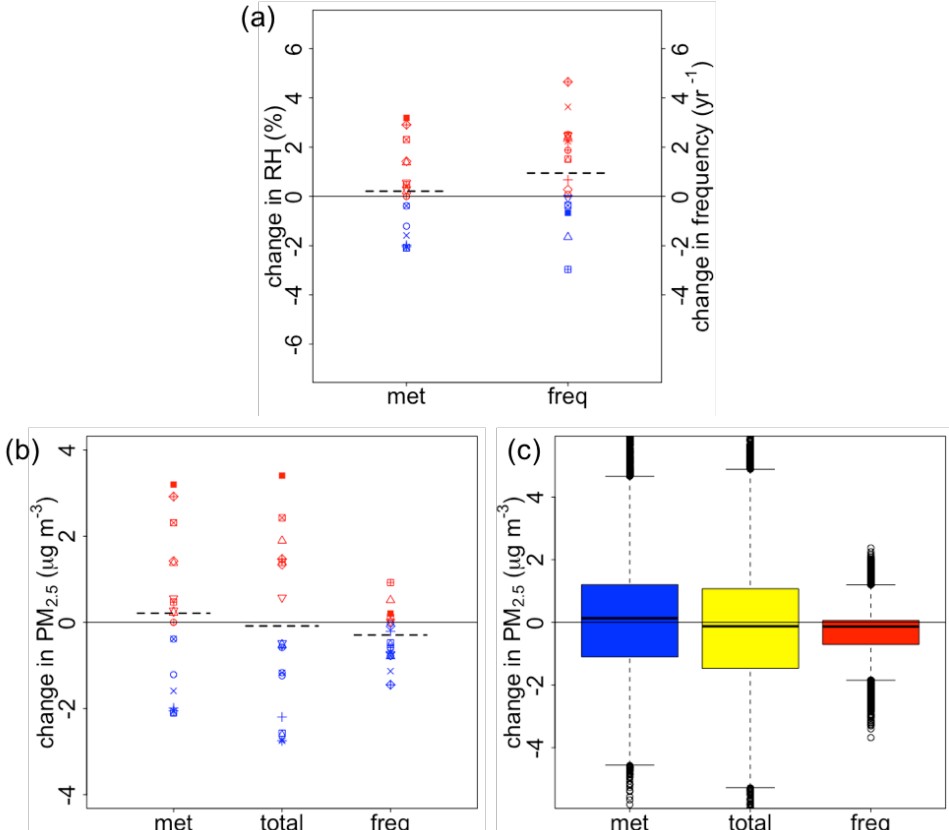

**Fig. 8.** Projected changes in PM$_{2.5}$ from 2000–2050, as calculated from meteorological output from the CMIP5 model ensemble. (a) Future projection of relative humidity and frequency of springtime Siberian high as computed by spectral analysis of principal component time series. (b) Change in PM$_{2.5}$ from 2000–2050 as computed by 15 models (in μg m$^{-3}$). (c) Monte-Carlo simulation of uncertainties of PM$_{2.5}$ projection (in μg m$^{-3}$). Dashed lines indicates the mean of the changes, red dots indicate positive changes and blue dot negative changes. The label "met" indicates changes associated with RH, "freq" indicates changes associated with frequency of the Siberian high, and "total" denotes the sum of the two.