# Peer review of "Synoptic meteorological modes of variability for fine particulate matter (PM2.5) air quality in major metropolitan regions of China"

_Atmospheric Chemistry and Physics, 2017_

## Referee Comment (RC1) · Anonymous Referee #1 · 16 Nov 2017

Leung et al. present a detailed study of meteorological drivers of present day PM2.5 seasonal and inter-annual variability in China. The authors leverage multiple data sets and sources including: meteorological reanalysis, observed PM2.5, satellite-derived PM2.5, and the CMIP5 ensemble of models. Their analysis examines the relationship between PM2.5 and meteorology in several regions of China. PM2.5-meteorology relationships derived from present day conditions are used to project how future climate may affect PM2.5.

Major comments:

1. The need for different data sets is explained (surface observed PM2.5 is limited in

terms of how long it has been collected, satellite PM2.5 is available for a longer time period, but only on an annual average basis, etc), but it's not clear to what degree using the different data sets leads to the same or differing conclusions. For example, how well do the meteorological principle components explain surface PM2.5 (if annually aggregated) vs satellite PM2.5?

2. Can the ability to project the effects of future conditions be strengthened? The r-squared of the PM2.5-meteorological model is 0.31 indicating it explains 31% of the interannual variability in PM2.5. The factor explaining the most variability is the Siberian High followed by RH. Could the r-squared be meaningfully increased by using all available meteorological variables? While the authors point out that meteorological variables co-vary, does that matter when trying to determine changes over a period of decades?

General comments:

1. Page 4, line 21: How were the relevant meteorological variables determined? Was there data on other variables that was not used or was this all that was available?

2. Can you clarify the relationship between the meteorological drivers and PM2.5 and whether information about one influences the determination of the other. Specifically:

2a. Near page 8: Meteorological principle components are determined without information on PM2.5 and then a regression is performed to determine how the meteorological PCs are related to PM2.5. Did the authors consider performing the principle component analysis with PM2.5 information (such as using PM2.5 as a variable in the PCA)?

2b. Near page 10: Can you comment on whether the synoptic modes represent purely meteorological features vs. any emission driven influences? For example, is there a mode that might cause increased emissions due to cold temperatures and increased home heating requirements thus increasing PM2.5 for reasons not driven by meteorol-

ogy?

3. Page 13, line 10: Returning to your earlier objective regarding separating synoptic features vs individual meteorological drivers, what remaining variability can be explained by meteorological variables that are not related to synoptic patterns?

4. Figure 2: Explain more how correlation with wind direction was determined. There were separate indicators for east/west and north/south (X7, X8) but panel (g) indicates one metric for direction.

Technical corrections:

1. Page 4, line 27: Add equation for deseasonalization and normalization.

2. Page 6, line 1, is subscript "k" needed on x bar?

3. Page 12, line 6: Figure 8a seems to have more than 3 models with negative changes in frequency in contrast to the text.

4. Page 12, line 19: What baseline PM2.5 values are the changes referenced to?

5. Figure 4: color bar indicates mm/day, not temperature.

6. Table 2: indicate observed PM2.5 or satellite-derived PM2.5.

---

## Referee Comment (RC2) · Anonymous Referee #2 · 22 Dec 2017

The manuscript presents an interesting study on the associations between PM2.5 pollution and meteorology over China, as well as projections of climate-induced impacts on PM2.5 pollution by 2050. This is a well-written manuscript that build on years of work and prior studies. It presents a significant scientific contribution from a methodological perspective and provides insight on the meteorological drivers of PM2.5 pollution over China. I recommend publication after the comments described below are addressed:

- The study mainly consists of 3 analyses: Correlations between PM2.5 and meteorological variables; (2) PCA/PCR for dominant modes for PM2.5 variability; (3) a regression model for climate change impact on PM2.5. The three analyses could be better

[Figure]

connected between them. While conclusions are drawn from each, I see do not see shared findings and limited connections among them. For example, the correlation analysis shows the strongest correlations between PM2.5 and temperature. However, temperature is not one of the predictors considered in the regression model used in Section 5. I assume this may be due to the correlation between temperature and relative humidity, and the prescence of temperature in the PC used, however this is not discussed. I believe an improved description of the motivation for each approach, as well as what unique and shared information can be drawn from them would be beneficial.

- The authors' analysis of climate change impacts on PM2.5 is unconvincing. The multimodel ensemble reflects a very high uncertainty in projections of relative humidity and frequency of springtime among the CMIP5 simulations. A regression model based only 2 predictors with a R2=0.31, which projects a very small change in PM2.5 (-0.13±2.10) for which even the sign of change is highly uncertain, is used to draw the conclusion that there will be a more likely than not decrease in PM2.5 pollution due to climate change in the region. I also found the description and results interpretation of the Monte Carlo analysis to be incomplete. Given the limitations of the modeling approach and the high uncertainties encountered, I felt the analysis described in section 5 does not truly suggest a climate benefit for PM2.5 in the BTH region, but rather demonstrates that the evidence of a climate-induced impact is inconclusive. This would agree with other global and US studies that have shown how challenging it may be to robustly project a climate-induce change on regional air quality by midcentury under natural variability and the large uncertainties in climate projections.

- Although the manuscript is well-written, the introduction includes some odd wording and grammatical mistakes. I recommend a careful review of this section by a native English speaker.

- Page 2, line 3-4: "attributed" is used incorrectly

[Figure]

- Page 2, line 3-4: "attribute" is used incorrectly

- The 2nd paragraph in the introduction should probably be combined with the first. As is, the second paragraph seems repetitive and oddly placed.

- Page 5, line 2: 1497 monitors seems like a large number of monitors; are all plotted in figure 1?

- Page 5, line 19-22: The description of how the AOD-based PM2.5 concentration fields are derived is unclear (e.g. what model simulation?). Improve this description.

- Section 3 and figure 2. The authors mention that PM2.5 sites in much of southwestern China are relatively sparse and these regions are excluded from the analysis. However, it seems that in correlation analysis in section 3, the entire country is considered. Figure 2 does not indicate a difference between grid cells that were excluded or those that are included but have correlation coefficients near 0. I would recommend clearing indicating cells in excluded regions (e.g. coloring them gray) and not drawing any conclusions from those locations.

- Section 3 and figure 2: Some of the correlations between PM2.5 and meteorological variables reported and mapped are very small, yet the authors still draw conclusions above how some of these may drive PM2.5 concentrations. For example, the correlation coefficients for precipitation, pressure tendency and windspeed tend to be below 0.2. Is it still appropriate to draw about the interactions between these variables and PM2.5 concentrations if the explain <5% of the variance?

- Following the previous comment, for example, why would the correlation between precipitation and PM2.5 be positive over parts of central and western China?

- Figure 2: Remove the statistically insignificant vectors from panel (g).

- Page 6, line 35: I am not sure I clearly see the 2 divergent wind patterns on the map, and I am not sure the author's conclusion that "wind transports pollutant from source regions to the peripheries" is substantiated. Which are the sources in these 2

locations?

- Section 5: The regression model explains about 30% of the variance in annual PM2.5 in the BTH region. Is this correlation string enough to draw conclusions about the climate "benefit" under the RCP scenario? I recommend discussing how this meteorology-driven climate impact is expected to compare with other drivers of PM2.5 change along this emissions pathway.

- Page 11, line 33: If the correlation with RH is statistically insignificant, do not report the value.

- Page 23, line 24: Change Monta to Monte

---

## Author Comment (AC1) · 4 Mar 2018

We thank the reviewers for their careful examinations and thoughtful comments. Our point-by-point responses are provided in the supplement.

Please also note the supplement to this comment: https://www.atmos-chem-phys-discuss.net/acp-2017-916/acp-2017-916-AC1-supplement.pdf

---

## Author Response (AR1)

Responses to Reviewers on "Synoptic meteorological modes of variability for fine particulate matter (PM2.5) air quality in major metropolitan regions of China" by Danny M. Leung et al. (MS No: acp-2017-916)

We thank the reviewers for their careful examinations and thoughtful comments. Our point-by-point responses are provided below. The reviewers' comments are *italicized*, our new/modified text is highlighted in **bold**.

Response to Referee #1

*Major comments: 1. The need for different data sets is explained (surface observed PM2.5 is limited in terms of how long it has been collected, satellite PM2.5 is available for a longer time period, but only on an annual average basis, etc), but it's not clear to what degree using the different data sets leads to the same or differing conclusions. For example, how well do the meteorological principle components explain surface PM2.5 (if annually aggregated) vs satellite PM2.5?*

For now we only have site data from June 2014 to May 2017, so that we can only aggregate two years of annual mean site PM$_{2.5}$ and could not yield a statistical relationship with the annual frequency of meteorological principal components. We plot below the gridded annual mean site PM$_{2.5}$ vs satellite PM$_{2.5}$ for year 2015, which shows that site data and satellite data correspond very well with each other with a correlation coefficient of 0.79, although satellite estimates systematically show slightly smaller values than site observations. This plot is now added as a panel to Fig. S1. Therefore, it is likely that when a longer (e.g., decadal) record of nationwide site observations of PM$_{2.5}$ are available in the future, annually aggregated site PM$_{2.5}$ should in general yield relationships with annual meteorological data that are similar to satellite-derived PM$_{2.5}$, but such an expectation can only be justified when a decadal, nationwide high-coverage site data record is available for China.

[Figure]

**Fig. S1b.** Annual mean satellite-derived PM$_{2.5}$ concentrations (µg m$^{-3}$) from van Donkelaar et al. (2016) versus gridded annual mean site PM$_{2.5}$ concentrations (µg m$^{-3}$) from the Chinese Ministry of Environmental Protection (MEP, http://pm25.in), for year 2015. The blue line indicates the fitted line using reduced major axis (RMA) regression with an $R^2$ value of 0.63, and the dashed line indicates the 1:1 line.

*2. Can the ability to project the effects of future conditions be strengthened? The r-squared of the PM2.5-meteorological model is 0.31 indicating it explains 31% of the interannual variability in PM2.5. The factor explaining the most variability is the Siberian High followed by RH. Could the r-squared be meaningfully increased by using all available meteorological variables? While the authors point out that meteorological variables co-vary, does that matter when trying to determine changes over a period of decades?*

We use adjusted $R^2$ value instead of multiple $R^2$ value for the MLR model, which gives penalty when adding more predictors into the MLR model to avoid overfitting. Even when the meteorological variables are not correlated on an interannual timescale, adding more of them in the MLR will not necessarily lead to higher adjusted $R^2$ values. In general, we are cautious about multicollinearity arising from co-varying meteorological variables, which generally enhances the standard errors of regression statistics and is thus deemed not desirable for prediction purposes. For BTH case, adding precipitation helps increase the adjusted $R^2$ value to 47%, but precipitation has a strong correlation of $r = 0.84$ with RH and thus was banned by the selection algorithm. In this case, RH is believed to be the true driver of $PM_{2.5}$, while precipitation is correlated with high RH only. Therefore, adding co-varying meteorological fields in projecting future $PM_{2.5}$ may not only double-count the same practical effect, but also increase the risk of introducing more projection errors due to multicollinearity itself and due to CMIP5 intermodel differences, leading to even higher $PM_{2.5}$ projection uncertainty. We also lengthen our discussion there:

p.12 line 7: "… Variables that lead to a large variance inflation factor (>2) are also excluded to avoid the issue of multicollinearity, **which often leads to higher imprecision of regression estimates**. Typically the forward selection algorithm…"

*General comments: 1. Page 4, line 21: How were the relevant meteorological variables determined? Was there data on other variables that was not used or was this all that was available?*

We follow Tai et al. (2012a, b) to select meteorological variables for PCA/PCR. There are more meteorological variables available in NCEP/NCAR Reanalysis 1, but we have chosen all variables that are directly related to synoptic circulation patterns as shown in previous studies. For clarity, we now add in the main text Sect. 2:

"… **Following Tai et al. (2012a, b), eight** meteorological variables are considered here (Table 1), …"

*2. Can you clarify the relationship between the meteorological drivers and PM2.5 and whether information about one influences the determination of the other. Specifically:*
*2a. Near page 8: Meteorological principle components are determined without information on PM2.5 and then a regression is performed to determine how the meteorological PCs are related to PM2.5. Did the authors consider performing the principle component analysis with PM2.5 information (such as using PM2.5 as a variable in the PCA)?*

Yes, we performed PCA with daily mean $PM_{2.5}$ included as one variable as well, and the result was similar to that of the PCR stated in this paper. We included the PCR result in order to follow the approach of Tai et al. (2012a, b), which is preferable to including $PM_{2.5}$ in the PCA directly because it also gives the relevant diagnostic statistics such as $R^2$ values, $p$-values, and most importantly, regression sums of squares that help quantify specifically the variability of $PM_{2.5}$ explained by meteorology (instead of the variability of the entire meteorology-$PM_{2.5}$ multidimensional space). We now say so at the end of Sect. 3:

p.7 line 34: "… We follow Tai et al. (2012a), and regress daily $PM_{2.5}$ on the resulting principal component (PC) time series to identify the dominant synoptic drivers of $PM_{2.5}$ variability. **Their approach is particularly useful in that it enables the quantification of the fraction of $PM_{2.5}$ variability that can be explained by synoptic meteorological regimes.**"

*2b. Near page 10: Can you comment on whether the synoptic modes represent purely meteorological features vs. any emission driven influences? For example, is there a mode that might cause increased emissions due to cold temperatures and increased home heating requirements thus increasing PM2.5 for reasons not driven by meteorology?*

We agree with the reviewer that covariation of emissions with meteorology may in part contribute to the observed $PM_{2.5}$ variability. Since this paper focuses on meteorological effects on $PM_{2.5}$ variability, we did not incorporate any emission data in the analysis. Daily emissions may partly co-vary with temperature because of indoor warming, as the reviewer has well noted, but our PCA results also show that cold fronts from the Siberian high (dominant PC of BTH case) are the main drivers of temperature and $PM_{2.5}$ variability. It is possible that during the arrivals of cold fronts, an increase in indoor warming might increase $PM_{2.5}$ that partly counteracts the cold-frontal ventilation effect and reduces the overall sensitivity of $PM_{2.5}$ to the dominant PC. However, we cannot infer how much it would have contributed to the overall sensitivity unless we have daily emission data for PCA altogether with meteorological variables so that the meteorology-induced emission variability can be incorporated in some other independent PC(s). This issue is now discussed in Sect. 4:

p.9 line 16: "… as well as information on the values of SSR/SST and $\beta$, is included in the supplementary materials. **We also note that in our interpretation, we focus only on the physical effects of meteorological phenomena. Non-physical drivers such as anthropogenic emissions can be correlated with meteorology to some extents (e.g., cold weather leading to higher emissions from heating); such effects, if any, would be encapsulated in the statistical model, but are difficult to diagnose explicitly due to a lack of corresponding data.**"

*3. Page 13, line 10: Returning to your earlier objective regarding separating synoptic features vs individual meteorological drivers, what remaining variability can be explained by meteorological variables that are not related to synoptic patterns?*

Here we take BTH as an example. Regressing PM$_{2.5}$ on only the annually dominant PC can yield an $R^2$ value of 0.36. Regressing PM$_{2.5}$ on the dominant PC and all local meteorological variables yields an $R^2$ value of 0.43.

We add this result into the paper for all four regions. For instance, p.9 line 35: "… PM$_{2.5}$ concentration decreases by nearly 240 μg m$^{-3}$ over 29 to 31 Dec (Fig. 3a). **Regressing PM$_{2.5}$ on all eight individual meteorological variables yields an $R^2$ value of 43%, indicating that local meteorology only contributes to an extra 7% of the PM$_{2.5}$ variability in addition to that already explained by synoptic circulation.** In addition to cold fronts from the Siberian high, …", and likewise for all four regions.

*4. Figure 2: Explain more how correlation with wind direction was determined. There were separate indicators for east/west and north/south (X7, X8) but panel (g) indicates one metric for direction.*

To get the vector plot for wind-direction correlation, we compute the correlation maps for $X_7$ and $X_8$ as with the other meteorological variables, and then convert them into vector plots with positive values pointing toward east and north, respectively. Adding up the two vector plots gives the final plot in Fig. 2g, which we find the most helpful way to visualize the effect of wind directions. We now add the detailed description in Fig. 2:

"… **Fig. 2g is plotted by finding the vector sums of the correlation coefficients for $X_7$ and $X_8$, with positive correlations pointing eastward and northward, respectively. The direction of the vector sum indicates the prevalent wind direction when PM$_{2.5}$ has a positive anomaly.**"

*Technical corrections: 1. Page 4, line 27: Add equation for deseasonalization and normalization.*

We add in the main text near p.4, line 28:

"…meteorological data except $X_5$, $X_7$ and $X_8$ are deseasonalized and detrended by subtracting the corresponding centered 30-day moving averages from the original data to focus on day-to-day, synoptic-scale variability. **Specifically, for a meteorological variable $X_k$ in any grid, the deseasonalized meteorology $\widetilde{X}_k$ is calculated as follows:**

$$\widetilde{X}_k(t) = X_k(t) - \frac{1}{31}\sum_{n=t-15}^{t+15} X_k(n) \tag{1}$$

**The deseasonalized and detrended data are also normalized to their standard deviations to yield zero means and unit variances:**

$$\widehat{X}_k(t) = \frac{\widetilde{X}_k(t) - \overline{\widetilde{X}_k}}{s_{\widetilde{X}_k}} \tag{2}$$

**where $\widehat{X}_k(t)$ represents the normalized meteorological time series, $\overline{\widetilde{X}_k}$ and $s_{\widetilde{X}_k}$ are the mean and standard deviation of the deseasonalized time series, respectively.**"

For clarity, we also change the text in Sect. 4, first paragraph:

"… The resulting time series for each region are then decomposed to produce the PC time series ($U_j = U_1, …, U_8$):

$$U_j(t) = \sum_{k=1}^{8} \alpha_{kj}\widehat{X}_k(t) = \sum_{k=1}^{8} \alpha_{kj}\frac{\left[\widetilde{X}_k(t)-\overline{\widetilde{X}_k}\right]}{s_{\widetilde{X}_k}} \tag{4}$$

**where $\widetilde{X}_k$ represents the deseasonalized regionally averaged meteorological fields in Table 1, $\overline{\widetilde{X}_k}$ and $s_{\widetilde{X}_k}$ are the temporal mean and standard deviation of $\widetilde{X}_k$, $\widehat{X}_k$ is the normalized value of $\widetilde{X}_k$,** and $\alpha_{kj}$ are the elements of the transformation matrix (i.e., eigenvector or empirical orthogonal function, EOF) of PCA. …"

*2. Page 6, line 1, is subscript "k" needed on x bar?*

Subscript "$k$" is needed because it represents the $k^{th}$ meteorological variable, and it is particularly useful to distinguish from the index $j$ used in PCA.

*3. Page 12, line 6: Figure 8a seems to have more than 3 models with negative changes in frequency in contrast to the text.*

Upon further checking, there are five out of 15 models with negative changes in frequency. We change the text in p.15 line 29, as follows:

"… under the RCP8.5 scenario. **Ten** out of 15 models show an increase in the frequency of strength fluctuation of the Siberian high **with an ensemble mean of 1.46 yr$^{-1}$. Nine out of 15 models show a statistically insignificant change in future RH.** Intermodel differences in the projected changes…"

*4. Page 12, line 19: What baseline PM2.5 values are the changes referenced to?*

The $PM_{2.5}$ changes according to our statistical model is with reference to the baseline value for the decade of 2000s, namely, 57.2 µg m$^{-3}$. We now add in the main text near p.16 line 7:

"Figure 8c and 8d shows the corresponding **future** $PM_{2.5}$ changes **from the baseline value of 57.2 µg m$^{-3}$ in the 2000s.** Across model results, …"

*5. Figure 4: color bar indicates mm/day, not temperature.*

Figure 4 is revised as suggested.

*6. Table 2: indicate observed PM2.5 or satellite-derived PM2.5.*

We now change the description of Table 2 as:

"Regression model that explains **interannual variability of satellite-derived PM$_{2.5}$ in Beijing-Tianjin-Hebei (BTH)**."

Response to Referee #2

*- The study mainly consists of 3 analyses: Correlations between PM2.5 and meteorological variables; (2) PCA/PCR for dominant modes for PM2.5 variability; (3) a regression model for climate change impact on PM2.5. The three analyses could be better connected between them. While conclusions are drawn from each, I see do not see shared findings and limited connections among them. For example, the correlation analysis shows the strongest correlations between PM2.5 and temperature. However, temperature is not one of the predictors considered in the regression model used in Section 5. I assume this may be due to the correlation between temperature and relative humidity, and the presence of temperature in the PC used, however this is not discussed. I believe an improved description of the motivation for each approach, as well as what unique and shared information can be drawn from them would be beneficial.*

We mentioned in the text (paragraph 3 of Sect. 6) that we did not see significant correlation between annual mean temperature and PM$_{2.5}$ ($r = 0.18$), indicating temperature is not the main driver of interannual PM$_{2.5}$ variability. To improve the connection between analysis (1) and (3) and in response to the reviewer's concern, we now extend the discussion in paragraph 3 of Sect. 5:

"… Annual mean RH has a positive correlation with PM$_{2.5}$ ($r = 0.49$), which is consistent with Sect. 3 where we found higher RH coinciding with higher PM$_{2.5}$ on the daily timescale. Adding RH helps explain an additional 9% interannual PM$_{2.5}$ variability, and the two predictors in total give an adjusted $R^2$ value of 31%. … **Although temperature has a strong daily correlation of $r = 0.6$ with PM$_{2.5}$ in the correlation analysis in Sect. 3, annual mean temperature does not appear to correlate significantly with annual mean PM$_{2.5}$ ($r = 0.18$) and was not selected by the forward selection algorithm. Annual mean temperature also has a weak correlation with springtime Siberian high fluctuation frequency ($r = -0.25$), which indicates that more frequent synoptic fluctuations have only little bearing on annual mean temperature, and that the strong daily PM$_{2.5}$-temperature covariation is mostly a manifestation of synoptic influence. Other annual mean local meteorological variables all have insignificant correlations with annual mean PM$_{2.5}$.**"

Moreover, to better connect between analysis (1) and (2), we show that the percentage of PM$_{2.5}$ variability explained by local meteorology that is independent of synoptic meteorological modes can be inferred from the PCR model. This answers the question of how much PM$_{2.5}$ variability explained by local meteorological variables in Sect. 3 is from synoptic effects. In p.9, line 4:

"… Any PC which has an SSR/SST more than half of that of the dominant PC in a given season is also recognized as an important PC for that region. **The total**

**percentage of PM$_{2.5}$ variability explained by the $K$ dominant synoptic modes in a region can be written as:**

$$R^2_{\text{synoptic}} = \Sigma_j^K R^2_{\text{synoptic},j} \qquad \qquad \text{(6b)}$$

**The PCR model also allows us to separate between synoptically driven and locally driven PM$_{2.5}$ variability from the total meteorologically driven PM$_{2.5}$ variability. Regressing PM$_{2.5}$ using all eight individual meteorological variables yields a total $R^2$ value, which entails both synoptically and locally driven PM$_{2.5}$ variability, as discussed in Sect. 3. Using $R^2$ and $R^2_{\text{synoptic}}$ from the PCR model, we can infer the variability explained by local meteorology alone unrelated to synoptic modes, using:**

$$R^2_{\text{local}} = R^2 - R^2_{\text{synoptic}} \qquad \qquad \text{(6c)}$$

**where $R^2_{\text{local}}$ indicates the overall locally driven PM$_{2.5}$ variability."**

The corresponding results are also included for all four regions, as explained above in our response to point 3 of the first reviewer.

*- The authors' analysis of climate change impacts on PM2.5 is unconvincing. The multimodel ensemble reflects a very high uncertainty in projections of relative humidity and frequency of springtime among the CMIP5 simulations. A regression model based only 2 predictors with a R2=0.31, which projects a very small change in PM2.5 (-0.13±2.10) for which even the sign of change is highly uncertain, is used to draw the conclusion that there will be a more likely than not decrease in PM2.5 pollution due to climate change in the region. I also found the description and results interpretation of the Monte Carlo analysis to be incomplete. Given the limitations of the modeling approach and the high uncertainties encountered, I felt the analysis described in section 5 does not truly suggest a climate benefit for PM2.5 in the BTH region, but rather demonstrates that the evidence of a climate-induced impact is inconclusive. This would agree with other global and US studies that have shown how challenging it may be to robustly project a climate-induce change on regional air quality by midcentury under natural variability and the large uncertainties in climate projections.*

We agree with the reviewer's opinion that our PM$_{2.5}$ prediction is small, with large uncertainty that comes more from the intermodal differences among climate projections than the statistical model. We found an adjusted $R^2$ value of 0.31 using two meteorological predictors, which we believe is moderately high, given that emission-driven interannual variability was also not considered in the model, and the robustness is on par with previous work on PM$_{2.5}$-meteorology relationships. The main source of uncertainty comes from the CMIP5 intermodel differences especially for future RH change, leading to high uncertainty of the RH-induced PM$_{2.5}$ projection. However, PM$_{2.5}$ change due to future change in Siberian high fluctuation alone is statistically significant ($p$-value = 0.046). To further improve the statistical robustness, we now introduce a weighting algorithm by Tebaldi et al. (2005) that compare the climatological means of present-day simulations from each of the CMIP5 models to those of reanalysis data, so that the ensemble mean is now a weighted mean that discounts underperforming models with respect to RH or cold frontal frequency, instead of a simple mean over all 15 models. The Monte-Carlo simulation of PM$_{2.5}$ changes before (left) and after (right) using the weighting algorithm is plotted below:

[Figure]

[Figure]

The frequency-induced PM$_{2.5}$ change increases in magnitude from $-0.34\pm0.63$ μg m$^{-3}$ to $-0.46\pm0.28$ μg m$^{-3}$, and the RH-induced PM$_{2.5}$ change goes from $+0.21\pm1.78$ μg m$^{-3}$ to $-0.21\pm1.44$ μg m$^{-3}$. The weighting algorithm does not significantly reduce the uncertainty of RH projection since the CMIP5 models have more or less equally poor performance on present-day simulations. In response to the reviewer's concern, we would maintain the conclusion of frequency-induced PM$_{2.5}$ change while deemphasizing the RH-induced change, and state that the RH-induced result is inconclusive due to the high uncertainty of CMIP5 RH projections. We now replace the original figures in Fig. 8 with the new results, and modify the statements in the last two paragraphs of Sect. 5 accordingly:

"**We find that 10 of the 15 models project an increase in this synoptic frequency (Fig. 8a). Based on the weighting algorithm to discount poorly performing models, we project an overall *very likely* (i.e., 90–100% likelihood according to Box TS.1 in Stocker et al., 2013), statistically significant increase (*p*-value = 0.0008) in the frequency of synoptic-scale fluctuation of the Siberian high by 1.46±0.39 yr$^{-1}$ by the 2050s (Fig. 8b). …**"

"**Figure 8c and 8d show the corresponding future PM$_{2.5}$ changes from the baseline value of 57.2 μg m$^{-3}$ in the 2000s. Across the model results, we find an overall PM$_{2.5}$ change by –2.10 to +3.19 μg m$^{-3}$ due to changing RH, and by –1.45 to 0.92 μg m$^{-3}$ due to changing synoptic frequency. From the Monte-Carlo sampling of the performance-weighted distribution of meteorological changes and uncertainties of statistical parameters, the RH-induced PM$_{2.5}$ change is –0.21±1.44 μg m$^{-3}$ (*p*-value = 0.58), and the frequency-induced PM$_{2.5}$ change is –0.46±0.28 μg m$^{-3}$ (*p*-value = 0.028, 97% likelihood) (Fig. 8d). While the RH-induced PM$_{2.5}$ change is statistically insignificant and its sign inconclusive, we show that the higher frequency of fluctuation in the Siberian high alone, through enhancing cold-frontal frequency, could lead to a *very likely* reduction in annual mean PM$_{2.5}$ and thus constitute a slight climate "benefit" for PM$_{2.5}$ air quality over BTH of China. We find that the greatest uncertainty stems from large** intermodel differences in the future projections of RH, **which are much larger than those in the synoptic frequency projections**. The regression coefficients have relatively moderate standard errors (Table 2) **and contribute only little to the overall projection uncertainty**."

We also modify the relevant discussion parts in paragraph 3, Sect. 6:

"… Intermodel differences in the projected changes in RH are much larger than that in synoptic frequency of fluctuation in the Siberian High, owing to the high inconsistency in future projections of atmospheric humidity, especially on a regional scale (IPCC, 2013). **We combine the ensemble projection of RH and synoptic frequency with the $PM_{2.5}$-to-climate sensitivity from our statistical model to project future $PM_{2.5}$ changes, with uncertainties quantified using a Monte Carlo approach. While the RH-induced $PM_{2.5}$ change is insignificant and inconclusive, we project for the 2050s a statistically significant and *very likely* (~97% likelihood) decrease in $PM_{2.5}$ of $-0.46\pm0.28$ µg m$^{-3}$ due to increasing frequency in the fluctuation of the Siberian high. The overall projection is inconclusive mostly due to the highly uncertain RH projections.** Our prediction is comparable in magnitude with other studies (e.g., Jiang et al., 2013), as well as future predictions done for the US (Shen et al., 2017; Tai et al., 2012b; Pye et al., 2009; Avise et al., 2009) and Europe (Juda-Rezler et al., 2012), **but much smaller in magnitude than the baseline value of 57.2 µg m$^{-3}$ in the 2000s, suggesting that the "climate benefit" from higher synoptic frequency is rather small especially in comparison with what emission control effort could do to curb $PM_{2.5}$ concentrations in China**. Jiang et al. (2013) projected changes of $PM_{2.5}$ over China due to climate change alone under IPCC A1B scenario, and the resulting change over BTH is about +1 µg m$^{-3}$ averaged annually. They attributed their predictions to: 1) changing precipitation that leads to a change in wet deposition; and 2) increasing temperature that results in more volatilization of nitrate and ammonium, **which differs from our conclusion that cold-frontal ventilation dominates the $PM_{2.5}$-temperature correlation and total $PM_{2.5}$ response.** Our statistical results…"

The abstract is also modified:

"…We apply the resulting $PM_{2.5}$-to-climate sensitivities to IPCC Coupled Model Intercomparison Project Phase 5 (CMIP5) climate projections to predict future $PM_{2.5}$ by 2050s due to climate change, and find a modest decrease of **~0.5 µg m$^{-3}$** in annual mean $PM_{2.5}$ in the BTH region **due to more frequent cold-frontal ventilation under the RCP8.5 future, representing a small "climate benefit", but the RH-induced $PM_{2.5}$ change is inconclusive due to the large intermodel differences in RH projections.**"

On the other hand, we now include a more detailed description of the weighting algorithm as well as the Monte-Carlo method in paragraph 4, Sect. 5:

"…We diagnose the 2000–2050 changes in the decadal averages of these variables and the median frequencies of the constructed PCs **(Fig. 8a). To obtain an ensemble mean and distribution of the meteorological changes (Fig. 8b), we apply the weighting algorithm of Tebaldi et al. (2005) to the CMIP5 model outputs, which can discount any poorly performing models yielding meteorology that diverges from the present-day observations (using NCEP/NCAR reanalysis data in this study), and that diverges too much from the weighted ensemble mean, by giving those models a lower weight in the calculation of the ensemble mean and distribution.**"

**"We combine the meteorological changes with the PM$_{2.5}$-to-climate sensitivities (i.e., regression coefficients in Table 2) to obtain an estimate for the 2000–2050 change in annual mean PM$_{2.5}$ due to climate change alone (Fig. 8c), according to the following formula:**

$$\Delta PM_{2.5} = \sum_i^N \frac{\partial PM_{2.5}}{\partial x_i} \Delta x_i \qquad\qquad (7)$$

**where $\Delta PM_{2.5}$ is the total PM$_{2.5}$ change due to climate change, *N* is the total number of predictors selected by the forward selection algorithm, and $\Delta x_i$ is the change of the *i*th predictor selected by the algorithm. Here we make the "stationarity" assumption that the PM$_{2.5}$-to-climate sensitivities, $\partial PM_{2.5}/\partial x_i$, remain unchanged in the near future.** We then use a Monte-Carlo approach to characterize the probability distribution and statistical significance of the changes in PM$_{2.5}$ concentration arising from the uncertainties of the regression coefficients in the MLR model, as well as from the differences in model physics among CMIP5 models. **Our approach involves repeated (>5000 times) sampling of regression coefficients of the MLR model from their distributions as parameterized by the means and standard errors in Table 2, along with the sampling of the performance-weighted ensemble distributions of meteorological changes from the Tebaldi et al. (2005) algorithm. The sampling distributions are aggregated in accordance with Eq. (7) to obtain the final distributions of PM$_{2.5}$ changes for each predictor and the sum of the two (Fig. 8d).**"

*- Although the manuscript is well-written, the introduction includes some odd wording and grammatical mistakes. I recommend a careful review of this section by a native English speaker.*
*- Page 2, line 3-4: "attributed" is used incorrectly*
*- Page 2, line 3-4: "attribute" is used incorrectly*
*- The 2nd paragraph in the introduction should probably be combined with the first. As is, the second paragraph seems repetitive and oddly placed.*

It is now revised as suggested. To avoid overly long paragraphs, part of the second paragraph is moved to the first. Now the first paragraph of the introduction reads:

"Air pollution caused by high surface concentrations of particulate matter (PM) and ozone in megacities are of utmost public health concern in China nowadays (Xu et al., 2013). **China has experienced deteriorating air quality since the 1990s due to rapid industrial and economic development. Episodes of haze and smog pollution with dangerous levels of fine particulate matter (PM$_{2.5}$, particles with an aerodynamic diameter of or less than 2.5 μm) are becoming more common in the most developed and highly populated city clusters in China (Chan et al., 2008; Zhang et al., 2007; Zhang et al., 2014). For example, annual mean PM$_{2.5}$ concentration in Beijing increased dramatically from 12 μg m$^{-3}$ in 1973 to 66 μg m$^{-3}$ in 2013 (Han et al., 2016), with an average growth rate of +0.7 μg m$^{-3}$ yr$^{-1}$ for the past four decades.** Outdoor air pollution in China alone has been shown to cause over one million premature deaths every year (Cohen et al., 2017). Many epidemiological studies have documented the harmful effects of PM$_{2.5}$ on cardiovascular and respiratory health (Cao et al., 2012; Krewski et al., 2009; Madaniyazi et al., 2015; Pope and Dockery, 2006). **Urban PM$_{2.5}$ originates from many sources including power plant, industry, vehicular emissions, road and soil**

**dust, biomass burning, and agricultural activities (Zhang et al., 2015), but the regional concentrations are also strongly dependent on pan-regional transport (e.g., Jiang et al., 2013) and ventilation by atmospheric circulation (e.g., Chen et al., 2008; Zhang et al., 2012; Zhang et al., 2016).**"

*- Page 5, line 2: 1497 monitors seems like a large number of monitors; are all plotted in figure 1?*

Yes, all monitors are plotted in Fig. 1. Many of them are concentrated over the metropolitan regions and are stacked on top of each other in the plot.

*- Page 5, line 19-22: The description of how the AOD-based PM2.5 concentration fields are derived is unclear (e.g. what model simulation?). Improve this description.*

We now extend the discussion in p.5, line 29:

"… Total column aerosol optical depth (AOD) retrievals from multiple satellite instruments **and model simulations, such as the MODerate resolution Imaging Spectroradiometer (MODIS), the Multiangle Imaging SpectroRadiometer (MISR), and the GEOS-Chem chemical transport model, were weighted by the ground-based AOD observations from the Aerosol Robotic Network (AERONET) sun photometers. The daily AOD and near-surface PM$_{2.5}$ were simulated by GEOS-Chem to obtain the AOD-PM$_{2.5}$ relationship, which were applied to the satellite AOD retrievals to yield weighted PM$_{2.5}$ concentrations. Annual mean values of PM$_{2.5}$ were computed** and then calibrated to ground-based PM$_{2.5}$ observations using the Global Geographically Weighted Regression (GWR) method (Brunsdon et al., 1996). Figure S1 shows…"

*- Section 3 and figure 2. The authors mention that PM2.5 sites in much of southwestern China are relatively sparse and these regions are excluded from the analysis. However, it seems that in correlation analysis in section 3, the entire country is considered. Figure 2 does not indicate a difference between grid cells that were excluded or those that are included but have correlation coefficients near 0. I would recommend clearing indicating cells in excluded regions (e.g. coloring them gray) and not drawing any conclusions from those locations.*

We understand the reviewer's concern, but we think it is still worth keeping those remote gridboxes colored for the readers' reference. Although we did not draw any conclusions from most of western China, colored gridboxes mean that the results are still statistically significant and credible. The method of interpolation assumes that the PM$_{2.5}$ level and temporal variability in a gridbox can be inferred from any sites within some search distance ($d_{max}$ = 500 km in this paper). If a gridbox contains no PM$_{2.5}$ sites but its gridded PM$_{2.5}$ still yields significant correlations, it should have captured the PM$_{2.5}$-meteorology relationships at least some sites in its neighboring gridbox, but interpretation would be more difficult. To be consistent, we prefer to keep these statistically significant correlation values in the maps.

*- Section 3 and figure 2: Some of the correlations between PM2.5 and meteorological variables reported and mapped are very small, yet the authors still draw conclusions above how some of these may drive PM2.5 concentrations. For example, the correlation coefficients for precipitation, pressure tendency and windspeed tend to be below 0.2. Is it still appropriate to draw about the interactions between these variables and PM2.5 concentrations if they explain <5% of the variance?*

We think it is appropriate to gain some meaningful physical insights from the correlation plots even when the correlation is below $|r| = 0.2$, since we define the term "statistical significant" to be correlation with $p$-value $\leq 0.05$ which correspond to 95% confidence level (equivalent to $|r| \geq 0.06$). In consideration of using 3 years (1096 days) of data to compute the correlations, we believe a correlation of 0.2 is enough to warrant physical interpretation.

*- Following the previous comment, for example, why would the correlation between precipitation and PM2.5 be positive over parts of central and western China?*

It is because precipitation is strongly correlated with RH; even a small amount of rainfall in central and western China can be enough to increase water content and encourage aerosol formation while having minimal effect on wet deposition. Since the true driver of aerosol is RH, the correlation map of precipitation is similar but weaker in magnitude than that of RH. We now extend the text in paragraph 3 of Sect. 3:

"… As can be seen in Fig. 2c, negative correlation of precipitation with $PM_{2.5}$ in southern China is very similar to that of RH in Fig. 2b, likely reflecting the association of high RH with precipitation (Fig. 2c) and onshore wind (Fig. 2f) which can facilitate $PM_{2.5}$ deposition or ventilation (Zhu et al., 2012). **Such a strong association between RH and precipitation may also explain the apparently positive correlation between precipitation and $PM_{2.5}$ in northern China, where RH-promoted aerosol formation is likely more important than wet deposition in the overall relationship.**"

*- Figure 2: Remove the statistically insignificant vectors from panel (g).*

We have now removed the statistically insignificant vectors with $p$-value > 0.05 (equivalent to vector length $|r| < 0.06$), and Fig. 2g now looks like this:

[Figure]

*- Page 6, line 35: I am not sure I clearly see the 2 divergent wind patterns on the map, and I am not sure the author's conclusion that "wind transports pollutant from source regions to the peripheries" is substantiated. Which are the sources in these 2 locations?*

Here we plot the mass divergence map according to the vector plot above, using the continuity equation in Eulerian form:

$$-\frac{\partial \rho}{\partial t} = \nabla \cdot (\rho \vec{V}) = \left( \frac{\rho_{i+1,j} u_{i+1,j} - \rho_{i-1,j} u_{i-1,j}}{2\Delta x} + \frac{\rho_{i,j+1} v_{i,j+1} - \rho_{i,j-1} v_{i,j-1}}{2\Delta y} \right)$$

where $\rho$ is the average mass concentration of $PM_{2.5}$, and $\vec{V} = (u, v)$ is the correlation vector in the above plot. For simplicity, we assume $\Delta x = \Delta y =$ constant everywhere. The map is shown below:

[Figure]

The map has a unit of µg m$^{-3}$ day$^{-1}$, indicating the mass flux (divergence is red). We can compare this map with the above vector plot, and see that the two strongest divergent patterns are over BTH and Xinjiang, where $PM_{2.5}$ is advected out of the grid cells. The two regions are surrounded by the light blue colors, indicating the peripheries are affected by pollution from the source regions. As we pointed out in the paper, anthropogenic emission is the main source of pollution over BTH, whereas dust emission is the main source over Taklimakan desert in Xinjiang. We now include these results in the supplement, which are referred to in paragraph 5 of Sect. 3:

"Figure 2g shows the correlation of wind direction with $PM_{2.5}$, in which arrow directions indicate wind directions associated with increasing $PM_{2.5}$. **The corresponding mass divergence map together with its calculation is shown in the supplement (Fig. S3).** For instance, $PM_{2.5}$ increases with southeasterly wind for the whole eastern and northeastern China with a correlation of $r = 0.3$ on average. …

*- Section 5: The regression model explains about 30% of the variance in annual PM2.5 in the BTH region. Is this correlation strong enough to draw conclusions about the climate "benefit" under the RCP scenario? I recommend discussing how this meteorology-driven climate impact is expected to compare with other drivers of PM2.5 change along this emissions pathway.*

This concern has mostly been addressed above (see our response to point (2) above). We now further add to paragraph 3 of Sect. 5:

"… Adding RH helps explain an additional 9% interannual $PM_{2.5}$ variability, and the two predictors in total give an adjusted $R^2$ value of 31%, **which represents a reasonably high value for a linear model, given that nonlinear $PM_{2.5}$-meteorology interactions and emission-driven $PM_{2.5}$ variability are not included in the model.** …"

*- Page 11, line 33: If the correlation with RH is statistically insignificant, do not report the value.*

We removed the numbers in the conclusion section, but kept the numbers in Sect. 5 for the readers' own comprehension.

*- Page 12, line 24: Change Monta to Monte*

(Page 12, line 24) Revised as suggested.